# Functional and structural characterization of interactions between opposite subunits in HCN pacemaker channels

Mahesh Kondapuram [1,5], Benedikt Frieg[2,5], Sezin Yüksel[1], Tina Schwabe[1], Christian Sattler [1], Marco Lelle[1], Andrea Schweinitz[1], Ralf Schmauder [1], Klaus Benndorf[1], Holger Gohlke [2,3,4 ✉] & Jana Kusch [1 ✉]

Hyperpolarization-activated and cyclic nucleotide (HCN) modulated channels are tetrameric cation channels. In each of the four subunits, the intracellular cyclic nucleotide-binding domain (CNBD) is coupled to the transmembrane domain via a helical structure, the C-linker. High-resolution channel structures suggest that the C-linker enables functionally relevant interactions with the opposite subunit, which might be critical for coupling the conformational changes in the CNBD to the channel pore. We combined mutagenesis, patch-clamp technique, confocal patch-clamp fluorometry, and molecular dynamics (MD) simulations to show that residue K464 of the C-linker is relevant for stabilizing the closed state of the mHCN2 channel by forming interactions with the opposite subunit. MD simulations revealed that in the K464E channel, a rotation of the intracellular domain relative to the channel pore is induced, which is similar to the cAMP-induced rotation, weakening the autoinhibitory effect of the unoccupied CL-CNBD region. We suggest that this CL-CNBD rotation is considerably involved in activation-induced affinity increase but only indirectly involved in gate modulation. The adopted poses shown herein are in excellent agreement with previous structural results.

[1] Universitätsklinikum Jena, Institut für Physiologie II, Jena, Germany. [2] John von Neumann-Institut für Computing (NIC), Jülich Supercomputing Centre (JSC), and Institut für Biologische Informationsprozesse (IBI-7: Strukturbiochemie), Forschungszentrum Jülich GmbH, Jülich, Germany. [3] Institut für Pharmazeutische und Medizinische Chemie, Heinrich-Heine-Universität Düsseldorf, Düsseldorf, Germany. [4] Institut für Bio- und Geowissenschaften (IBG-4: Bioinformatik), Forschungszentrum Jülich GmbH, Jülich, Germany. [5] These authors contributed equally: Mahesh Kondapuram, Benedikt Frieg. ✉email: gohlke@uni-duesseldorf.de; jana.kusch@med.uni-jena.de

HCN (Hyperpolarization-activated and cyclic nucleotide-modulated) channels are non-selective cation channels that mediate critical neuronal and cardiac processes, including the generation of electrical rhythmicity, synaptic plasticity, somatic sensibility (reviewed in ref. [1]), and shaping of cardiac action potentials[2]. Structurally, they belong to the superfamily of voltage-gated potassium channels. In contrast to most members of this family, however, HCN channels are gated by a dual mechanism, combining two stimuli: hyperpolarization and cyclic nucleotide-binding[3–5]. Hyperpolarizing voltage alone can activate the channel, whereas binding of cyclic nucleotides, such as cAMP or cGMP, has only a modulatory effect on activation: It shifts the steady-state activation relationship to more depolarized voltages, increases the maximum current amplitude, accelerates the activation kinetics, and decelerates the deactivation kinetics.

HCN channels are tetramers. Each of the four subunits consists of a membrane portion, formed by six transmembrane domains (S1 to S6), with a pore region between S5 and S6, and S4 as the central part of the voltage sensor. The intracellular N-terminus harbors an α-helical structure prior to S1, the HCN domain[6]. The intracellular C-terminus carries the cyclic nucleotide-binding domain (CNBD), which is connected to the S6 helix via another α-helical structure, the C-linker (CL)[7]. Several studies suggested an autoinhibitory effect of the unoccupied CNBD-CL portion, hindering a full activation of the channel. The binding of cyclic nucleotides to the CNBD releases autoinhibition and maximizes activation[8–10].

Different techniques, including electrophysiological approaches, fluorescence microscopy, and cryo-electron microscopy combined with mutagenesis were used to show intensive interactions between neighboring subunits: S4-S5 linker-CL interactions[6,11], CL-CL interactions[12–14], CNBD-CNBD interactions[6,12], and very recently interactions between the newly discovered HCN domain with the CNBD, the voltage-sensing domain, and the CL region[6,15,16].

In a recent study, employing mutated murine HCN2 (mHCN2) concatamers with a defined number and stoichiometry of functional and disabled binding sites, we showed that cAMP occupation of two subunits in *trans* position led to a deceleration of deactivation to a similar extent as in a fully occupied wild-type channel[17]. By contrast, cAMP occupation of two subunits in *cis* position did not show any decelerating effect[17]. These data raised the question of whether a direct interaction between opposite subunits is essential for channel gating. This suggestion is supported by the latest high-resolution structures of HCN channels suggesting that the C-linker enables functionally relevant interactions with the opposite subunit, which might also be critical for coupling the conformational changes in the CNBD to the channel pore[6,15,18].

Here, we address this question by combining mutagenesis, electrophysiology, confocal patch-clamp fluorometry (cPCF), and molecular dynamics (MD) simulations for studying the mHCN2 channel. We show that charge inversion introduced by the K464E variant at the hinge between A′- and B′-helix of the CL (also known as elbow structure[14]) promotes channel activation, most likely by interactions with the opposite subunit. Regarding the widely accepted gating mechanism proposing a rotation of the CL-CNBD relative to the channel pore to unwrap the S6 helix bundle at the intracellular side of the membrane[19,20], we suggest that the K464-mediated interactions hinder such a rotation. Conclusively, breaking these interactions by substituting K464 promotes a rotation of the CL-CNBD relative to the channel pore, thereby destabilizing the closed conformation.

## Results

### 3D HCN channel structure suggests proximity between opposite subunits.
Previous studies on subunit interactions during HCN channel gating focused on inter-subunit interactions between two adjacent subunits (e.g., refs. [12,21]). Visual inspection of the recently published 3D structure of the human HCN1 (hHCN1) channel[6] revealed potential interactions between opposite subunits. Since, to date, no 3D structure of a full-length HCN2 channel is available, we built homology models of the mHCN2 channel in *apo* and cAMP-bound form based on the 3D structure of the homologous hHCN1 (Fig. 1a; sequence identity 80%; the mHCN2 model was built for the sequence from L136 to D650) to predict interactions between nearby residues of opposite subunits that may be essential for channel gating.

We identified a hydrogen bond between the side-chain of K464, located in the elbow of the CL of subunit *i*, and the backbone carbonyl oxygen of M155, found in the second α-helix of the HCN domain (HCNb) of subunit $i + 2$ (Fig. 1a, b). K464 is highly conserved among the four mammalian HCN isoforms (HCN1 to HCN4) and spHCN from sea urchin sperm and was also found in some mammalian and invertebrate cyclic nucleotide-gated channels (Fig. 1c). Channels without a positive charge at that same position carry a positive charge (lysine or arginine) one or two positions adjacent to that (Fig. 1c), suggesting that charged interactions involving this region are likely relevant for channel function. M155 is conserved in HCN1, HCN2, and HCN4. HCN3 and spHCN carry the hydrophobic amino acids valine and leucine in that position, respectively (Fig. 1d). Together, from the structural and sequence analyses, the question arises whether the interactions between opposite subunits are crucial for the functional integrity of HCN channels.

### K464 is involved in autoinhibition and cAMP-triggered gating enhancement.
To study the function of the highly conserved K464, we constructed two mHCN2 channel mutants, K464E for charge reversal and K464A for charge neutralization. We performed patch-clamp experiments in the inside-out configuration using excised macropatches from *Xenopus laevis* oocytes and compared the gating of these two mutants with that of mHCN2 wild-type channels. Both mutants led to robust currents. The results are summarized in Fig. 2.

Channel activation was studied by applying voltage families from −70 to −150 mV in 10 mV increments as shown exemplarily for HCN2 channels in the left (cAMP-free) and middle panel (saturating [cAMP] of 10 μM) in Fig. 2a (for representative current traces of K464E and K464A see Supplementary Figure S1). Steady-state activation relationships were obtained from tail currents at a test pulse of −100 mV, following the varying activating pulses (Fig. 2b, e). The Boltzmann equation (Eq. (1)) was fitted to relative tail current amplitudes of individual recordings, yielding the voltage of half-maximal activation, $V_{1/2}$, and the effective gating charge, $z\delta$. As expected from the literature, in HCN2 wild-type channels, the application of cAMP led to a pronounced shift of the steady-state activation relationship to more depolarized voltages ($\Delta V_{1/2} = 17.9 \pm 1.1$ mV (Fig. 2b, h). Interestingly, the relationship for K464E in the absence of cAMP resembled the cAMP-saturated mHCN2 wild-type channel by causing a $V_{1/2}$ value of $-96.1 \pm 1.0$ mV compared to a $V_{1/2}$ value of $-100.6 \pm 1.3$ mV. Hence, adding saturating cAMP concentrations shifted the relationship by a minor but significant extent of $5.24 \pm 1.1$ mV to more depolarized voltages (Fig. 2b, h).

To evaluate the activation kinetics, we fitted an exponential function (Eq. (2)) to the time courses of activating currents, yielding the time constant of activation, $\tau_{act}$. The results for K464E and mHCN2 are shown in Fig. 2c, d. Because K464E was responsive to a different range of command voltages, $V_{command}$, compared to wild-type mHCN2, resulting in different activation

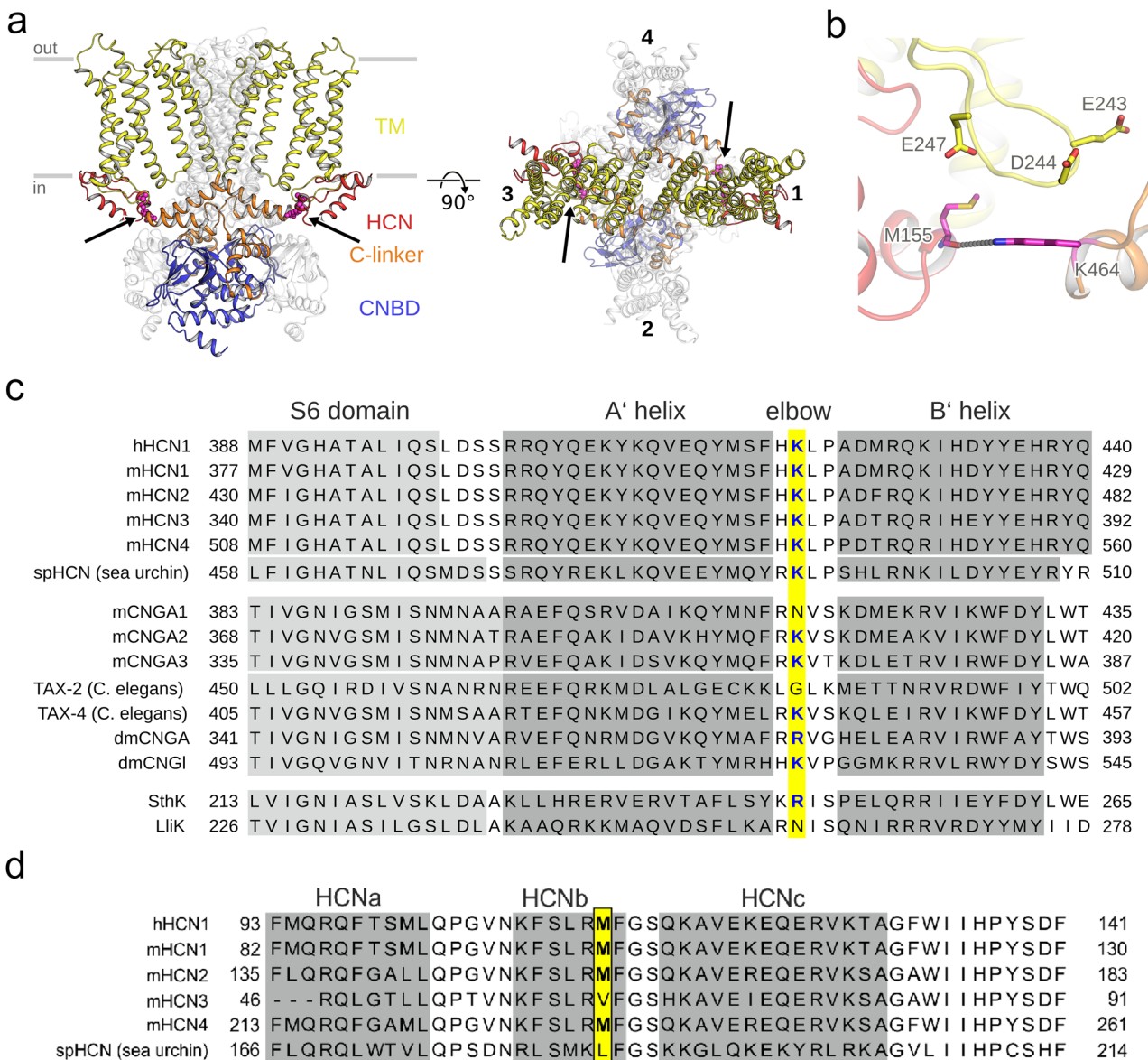

**Fig. 1 Identifying amino acids in potential opposite subunit interactions. a** Structural model of the homotetrameric mHCN2 channel (residues L136 to D650). The left panel shows a view from the side, and the right panel shows a view from the top. The gray bars depict the approximate location of the membrane bilayer. Two opposite subunits are either colored according to the domain organization with the HCN domain colored red, the transmembrane (TM) domain yellow, the CL domain orange, and the cyclic nucleotide-binding domain (CNBD) blue, or in full white. M155 and K464, residues suggested to form interactions between two opposite subunits, are depicted as magenta spheres and highlighted by an arrow. **b** A close-up view shows an overlay of the mHCN2 model. M155, E243, D244, E247, and K464 are depicted as sticks. The interaction between K464 and M155 is depicted as a gray dotted line. **c** Sequence alignment for CNBD channels. Positions equivalent to K464 in mHCN2 are highlighted in yellow. **d** Sequence alignment for HCN channels carrying an HCN domain. Positions equivalent to M155 in mHCN2 are highlighted in yellow.

states for each voltage, we decided to plot the activation time constants $\tau_{act}$ not only versus the command voltage (Fig. 2c), but additionally versus the normalized voltage $V_{command}/V_{1/2}$ (Fig. 2d). For control conditions in the absence of cAMP, and for high activation states in the presence of cAMP, there was no difference in activation kinetics for K464E and wild-type mHCN2. In contrast, K464E activation kinetics was accelerated compared to HCN2 activation kinetics for lower activation states when cAMP was bound. However, due to the limited time window of the activating pulse and the lower current amplitudes in this voltage range, those differences should be interpreted carefully. Together, the activation kinetics suggest that rate-limiting steps for channel activation were not substantially affected by the mutation K464E.

For studying deactivation, we used a protocol shown in the inset of panel Fig. 2g, with varied activating hyperpolarizing voltages and a deactivating voltage step to −30 mV. An exponential function (Eq. (2)) was fitted to the deactivation time courses, yielding the time constant of deactivation, $\tau_{deact}$. Because $\tau_{deact}$ was independent from the activating voltage pulse, only values obtained from −120 mV pulses are shown in Fig. 2d (for $\tau_{deact}$ values at the whole voltage range see Supplement Figure S2. The time constants of deactivation in both the presence and absence of saturating [cAMP] were similar to the time constants obtained from mHCN2 in the presence of saturating [cAMP].

The effective gating charge, $z\delta$, was significantly changed by mutating K464 (Fig. 2i), indicating an effect not only on cAMP-dependent but also on voltage-dependent gating.

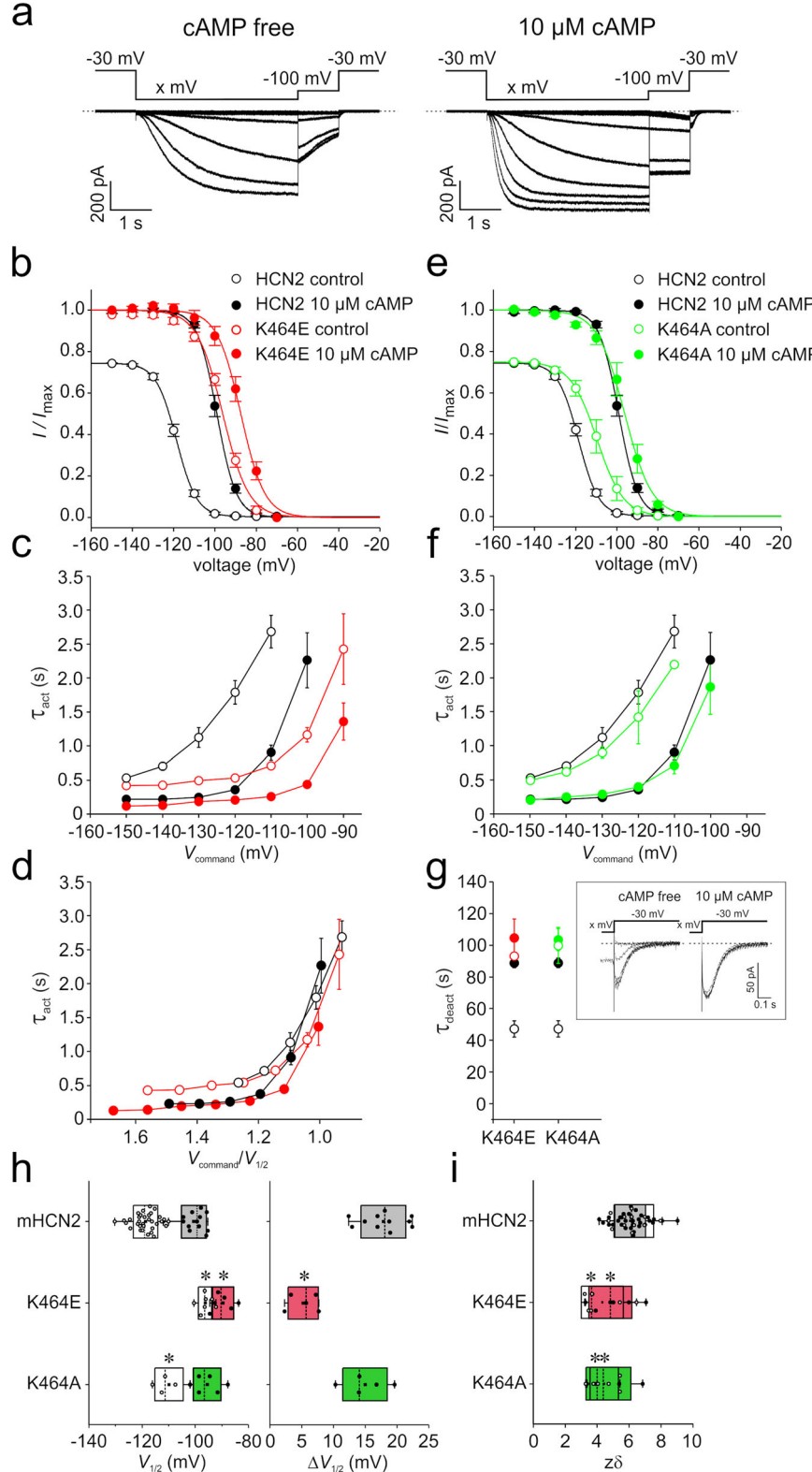

The same analysis was repeated for K464A (Fig. 2e–i). For this construct, both steady-state activation relationships, in the absence and in the presence of cAMP, were shifted only slightly to more depolarized voltages (Fig. 2e) with a reduced $\Delta V_{1/2}$ (15.0 ± 1.6 mV) compared to mHCN2 (Fig. 2h). Closer inspection of the activation and deactivation kinetics revealed that this slight stabilization of the open state is not caused by promoted activation (Fig. 2f) but by decelerated deactivation (Fig. 2g).

The effective gating charge, $z\delta$, was changed to a similar extent as for K464E (Fig. 2i).

In summary, the data show that in mHCN2 K464E destabilizes the closed (auto-inhibited) conformation, thereby affecting both the activation and the deactivation pathway. The similarity of the data obtained for wild type at saturating [cAMP] and K464E channels without cAMP led us to speculate that *apo* K464E and cAMP-bound channels behave similarly, but both differ from the

**Fig. 2 Voltage-dependent activation of K464 mutants at zero and saturating [cAMP] (10 μM). a** Exemplary current traces for mHCN2 activation. Left panel: protocol and representative traces for measuring steady-state activation and activation kinetics at zero [cAMP], right panel: as left panel but at saturating [cAMP] (10 μM). **b**, **e** Steady-state activation for K464E and K464A at zero and saturating [cAMP] in comparison to mHCN2, respectively. Solid lines indicate the result of a Boltzmann fit (equation 1) ($n = 5$ to 9). **c**, **f** Activation kinetics at zero and saturating [cAMP] for K464E, K464A, and mHCN2 ($n = 5$ to 9). **d** Activation kinetics for mHCN2 and K464E plotted versus normalized command voltage ($V_{command}/V_{1/2}$). **g** Deactivation kinetics at zero and saturating [cAMP] for K464E, K464A, and mHCN2 after −120 mV command voltage ($n = 5$ to 7). **h** Mean $V_{1/2}$ values and cAMP-induced shift of $\Delta V_{1/2}$ for all constructs ($n = 5$ to 33). **i** Effective gating charge $z\delta$ for all constructs. In all panels open symbols represent cAMP-free, filled symbols saturating cAMP conditions ($n = 5$ to 33). **h**, **i** Asterixes indicate significant difference between the respective mutant data and mHCHN2 data under the same cAMP condition. In all box plots dotted center lines represent median, box limits represent standard deviation, whiskers represent minimum and maximum values, cirles represent individual recordings, squares represent mean.

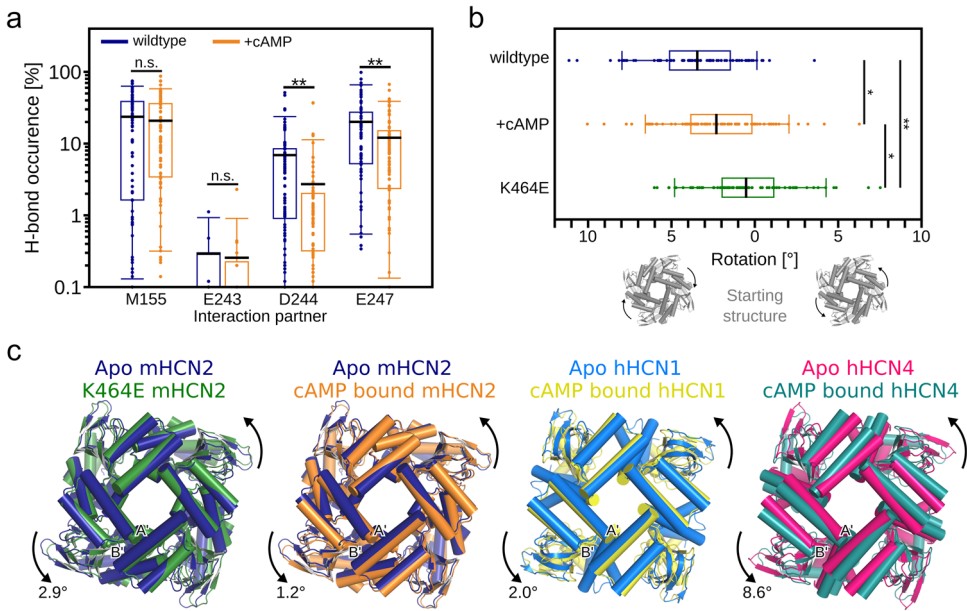

**Fig. 3 Analyses of MD simulations of the wild-type and K464E mHCN2 channel. a** Average occurrence frequency of hydrogen bond interactions between two opposite subunits involving K464. K464 resides on subunit $i$, and the interaction partners M150, E243, D244, and E247 reside on subunit $i + 2$. As to M155, we only considered the backbone oxygen as H-bond acceptor; for E243, D244, and E247, we only considered the side-chain oxygen atoms, as we considered these interactions more favorable compared to backbone interactions. **b** Average rotation angle relative to the channel pore in the starting structure. The direction of rotation is visualized by the scheme below the panels. **c** Overlay of CL-CNBDs after superimposing the pore regions. From left to right, the panels show the overlay of the average apo wild-type mHCN2 (dark blue) and the average K464E mHCN2 (green) or the average cAMP-bound wild-type mHCN2 (orange) from MD simulations, the overlay of the apo (light blue, PDB ID 5U6O[6]) and cAMP-bound (yellow, PDB ID 5U6P[6]) cryo-EM structures of hHCN1, and the overlay of the apo (magenta, PDB ID 6GYN[18] and cAMP-bound (dark cyan, PDB ID 6GYO[61] cryo-EM structures of hHCN4. Helices are shown as cylinders. The arrows indicate the direction of rotation relative to the respective apo structures. The labels depict the angle of rotation relative to the respective apo structures. A'- and B'-labels denote the first two helices of the CL. In **a** and **b**, the average values (black lines) were calculated individually for each of the four subunits and throughout 20 independent MD simulation replicas ($n = 80$) with the individual data points shown as circles. The boxes denote the range from the 25th to 75th percentile and include 50% of the data points. The whiskers denote the 5th to 95th percentile and include 90% of the data points ($p$-value by $t$-test; $*p < 0.05$; $**p < 0.01$; n.s. not significantly different).

*apo* wild-type channel. As K464A reveals only weak destabilizing properties on mHCN2, we assume that charged interactions likely mediate the destabilizing influence of K464E.

**K464E and cAMP induce CL-CNBD rotation relative to the *apo* channel.** To corroborate this hypothesis, we performed 20 independent replicas of unbiased MD simulations of 1 μs length each of *apo* wild-type mHCN2, wild-type mHCN2 bound to four cAMP molecules, and the *apo* K464E variant. All simulations were analyzed towards hydrogen bond and salt bridge interactions involving the residue K464, as shown in Fig. 1b.

In wild-type mHCN2, K464 forms a hydrogen bond with the backbone of M155 in the HCN domain (in 23.7% ± 2.5% of all conformations) (Fig. 3a), thereby bridging two opposite subunits. This hydrogen bond frequency is only marginally reduced upon

cAMP binding (n.s. rel. to *apo* wild type). By contrast, during the MD simulations of K464E, no hydrogen bonding with the backbone of M155 was recorded. The minimal distance between any of the side-chain oxygens in K464E to any backbone atom of M155 is usually >5 Å (Fig. S3). Thus, the direct interaction between two opposite subunits mediated by K464 is most likely completely lost in the K464E channel, thereby influencing the global channel structure.

The absence of K464-meditated interactions in the *apo* K464E mHCN2 promotes the CL-CNBD to adopt a conformation similar to that induced by cAMP binding to homologous HCN channels[6]. Throughout our simulations, the CL-CNBD rotates clockwise relative to the starting structure in all investigated mHCN2 channels. This rotation might be considered a relaxation of the mHCN2 channel due to starting from a homology model based on the hHCN1 structure. However, although starting from

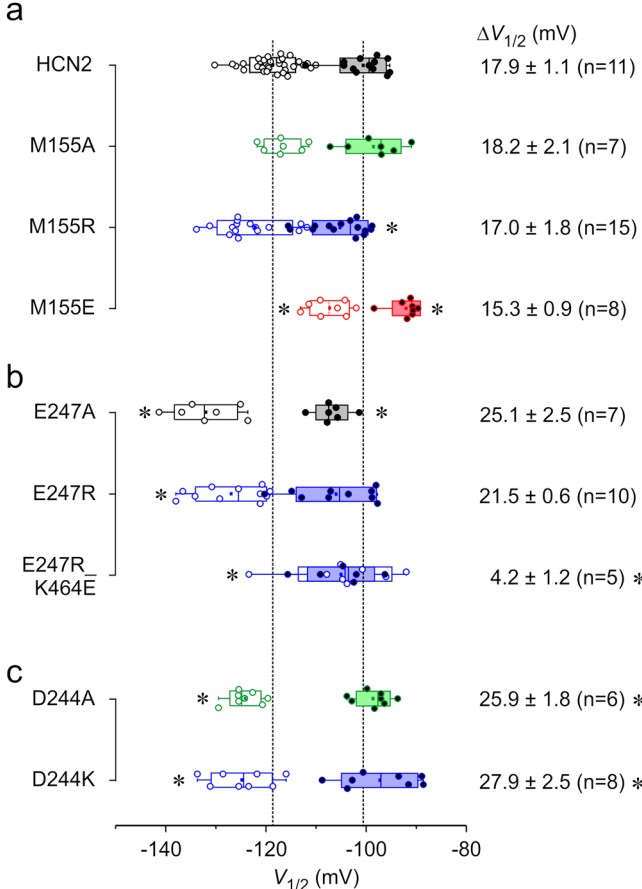

**Fig. 4 Voltage-dependent activation of potential interaction partners at zero and saturating [cAMP] (10 µM).** Box plots indicating mean $V_{1/2}$ values and SD for M155 constructs (**a**), E247 constructs (**b**), and D244 constructs (**c**). Open and filled symbols represent individual recordings for zero and saturating cAMP, respectively. Dotted center lines represent median, box limits represent standard deviation, whiskers represent minimum and maximum values, cirles represent individual recordings, squares represent mean. Numbers give the cAMP-induced shift of $V_{1/2}$, $\Delta V_{1/2} \pm$ SEM. $n$ indicates numbers of recordings, which were included in determining $V_{1/2}$. Only patches were included, in which both voltage families (with and without cAMP) could be recorded successfully. Asterisks indicate significant differences for comparison with mHCN2 at the respective cAMP condition (Student's $t$-test ($p < 0.05$)) or for the $\Delta V_{1/2}$ values.

the same channel conformation, the degree of rotation is significantly different between *apo* (3.45° ± 0.31°), cAMP-bound (2.31° ± 0.33°; $p < 0.05$ rel. to *apo* mHCN2), and K464E mHCN2 (0.52° ± 0.31°; $p < 0.01$ rel. to *apo* and cAMP-bound mHCN2) (Fig. 3b). Hence, the CL-CNDB in the K464E channel is displaced by ~2.9° relative to the *apo* wild-type channel (Fig. 3b), slightly larger than that observed in MD simulations of cAMP-bound wild-type mHCN2 and intermediate to rotation angles found in experimental hHCN1 and hHCN4 structures (Fig. 3c). Thus, our simulation data suggest that both cAMP binding and K464E substitution induce an anti-clockwise rotation of the CL-CNBD relative to the *apo* mHCN2 channel, similar to what is known from experimental studies on hHCN1 and hHCN4 (Fig. 3c). Still, as the hydrogen bond between M155 and K464 is insensitive to cAMP binding (Fig. 3a), one might assume that the rotation induced by K464E is not solely caused by the loss of a hydrogen bond to the backbone of M155. In line with this assumption, patch-clamp experiments revealed only small effects on the channel activation for the K464A mutant compared to the K464E

mutant, also suggesting that the gating behavior in K464E is not solely caused by the loss of a hydrogen bond to the backbone of M155, which would also occur in the case of K464A.

We next checked for interactions of K464 with the negatively charged residues E243, D244, and E247 on the S2/S3 linker (Fig. 1b), which are potential additional interaction partners of K464. In the *apo* wild-type channel, infrequent salt bridge interactions were found between K464 and E243 (in <0.5% of all conformations) or D244 (in 6.9% ± 1.1% of all conformations), whereas such interactions were found with E247 (in 20.1% ± 2.4% of all conformations; Fig. 3a). Thus, it seems reasonable to assume that repulsive electrostatic forces introduced by the K464E mutant contribute to the CL-CNBD rotation. Results from patch-clamp experiments support this assumption: The activation voltage is shifted towards more positive values in the K464E mutant relative to the wild-type channel, but to a lesser extent in K464A (Fig. 2e, h). Similarly, upon cAMP binding the interaction frequencies to D244 (in 2.7% ± 0.7% of all conformations; $p < 0.01$) and E247 (in 12.1% ± 1.6% of all conformations; $p < 0.01$) are significantly reduced (Fig. 3a).

Finally, to rule out that the effects on the CL-CNBD are solely artifacts introduced by homology modeling of the mHCN2, we conducted an additional set of MD simulations with the hHCN1 channel. The hHCN1 served as template for the homology modeling of the mHCN2 (sequence identity between mHCN2 and hHCN1 is 80% considering mHCN2 residues L136 to D650), and its atomic structure was previously resolved by cryo-EM to near-atomic resolution[6]. We again considered three different systems, *apo*, cAMP-bound, and K422E hHCN1. Note that K422 in hHCN1 is the homologous residue to K464 in mHCN2 (Fig. 1c). The simulation protocol and analyses were done identically as for mHCN2. The MD simulations revealed several interesting points, summarized in Fig. S4. First, the interaction pattern of K422 in hHCN1 is very similar to that of K464 in mHCN2. Second, K422E induces a significant anti-clockwise rotation by ~2.7° of the CL-CNBD relative to the *apo* hHCN1 channel. Third, cAMP binding to hHCN1 reduces interaction frequencies between K422 and negatively charged residues on the S2/S3-linker, similar to mHCN2, although the changes are not significant. This finding may be related to that the cAMP-induced rotation is not significantly different from zero. This observation may not be surprising, considering that the cAMP-induced rotation observed in cryo-EM structures of the hHCN1 is very small compared to other channels (Fig. 3c). Moreover, HCN1 channels show only a very weak response to cAMP[22], while HCN2[22–25] and HCN4 channels[24,26] respond strongly. Still, the histogram of rotation angles reveals that the CL-CNBD is more mobile after cAMP binding than in *apo* wild-type hHCN1, in line with the general notion that cAMP binding is associated with relieving CNBD-induced inhibition[10]. Thus, similar simulation data on hHCN1 suggest that the CL-CNBD rotation and the underlying structural interaction motive observed for mHCN2 are unlikely artifacts of using a homology model for HCN2.

**K464 acts via the backbone but not via side-chain interactions with M155 of the opposite subunit.** To test the hypothesis that the backbone rather than the side chain of M155 is involved in hydrogen bond interactions with K464, we mutated M155 to alanine, arginine, or glutamate and showed the effects of these substitutions on steady-state activation. If the backbone-side chain interaction is the predominant interaction between M155 and K464, we expect only minor effects, if any, due to these mutations. All three constructs led to functional channels and robust currents. The results are summarized in Fig. 4a and Fig. S5.

M155A showed no difference in the steady-state activation relationship compared to mHCN2, neither in the absence of cAMP nor in the presence of saturating [cAMP] (Fig. 4a, Fig. S5), resulting in a similar $\Delta V_{1/2}$. In M155R, the relationship in the presence of cAMP was slightly shifted to more hyperpolarized voltages while it was similar to mHCN2 in the absence of cAMP. However, the cAMP-induced $\Delta V_{1/2}$ was in the range of wild-type mHCN2. In M155E, both curves were shifted to more depolarized voltages, indicating an effect on voltage-dependent gating and stabilizing the open state. However, there was no significant effect on $\Delta V_{1/2}$; thus, the cAMP-dependent gating was not affected. The slopes for all three mutants, representing the effective gating charges, were not different from those of the wild-type channel (Fig. S5).

Neither shortening the side-chain at position 155 (M155A) nor adding a positive charge (M155R) to test for repulsive forces between position M155 and K464 affected the gating behavior similarly as did K464E, which supports our idea that the side chain at position 155 is not interacting directly with the side chain of K464. Moreover, in M155E, analysis of steady-state activation revealed an open state stabilization. If the introduced glutamate side chain interacted with the K464 side chain, most likely by forming a stabilizing salt bridge between the HCN domain and opposite C-linker, we would rather expect a stabilization of the closed state.

**Role of negatively charged residues in the S2-S3 linker for opposite subunit interactions**. As described above, E247 in the S2-S3 linker was identified as a potential interaction partner for K464 (besides M155) to form a salt bridge. Thus, we constructed the mutants E247A and E247R to study the role of this residue for channel gating. The data are summarized in Fig. 4b. For both channel variants, cAMP-induced gating is not affected and the shift of $V_{1/2}$ is similar to the shift shown for wild-type channels. However, voltage-induced gating was significantly affected: The steady-state activation relationships were shifted to more negative voltages for both mutants, indicating a stabilization of the closed state. From this, it can be concluded that changing the charge at position 247 in the S2-S3 linker, that way breaking a potential bond between E247 and K464, has no negative effect on the bond between K464 and M155. This result is further supported by the gating behavior of the mutant E247R_K464E. If a salt bridge between K464 and E247 mediated the function of K464, such a bond should be rescued in E247R_K464E, leading to a wild-type-like phenotype. However, in E247R_K464E, the cAMP dependence was still strongly reduced, like in K464E ($V_{1/2} = 4.2 \pm 1.2$ mV) (Fig. 4b).

In addition, our MD data showed a low probability of forming salt bridge interactions between K464 and D244 (Fig. 3a). We tested the role of D244 for channel gating by constructing D244A and D244K. In both cases, $V_{1/2}$ values were shifted to more negative values in the absence of cAMP, while there were no changes for cAMP-saturated constructs (Fig. 4c). Consequently, $\Delta V_{1/2}$ was significantly increased rather than decreased, as shown for K464E. These data indicate that changing the charge at position 244 in the S2-S3 loop, that way breaking a potential bond between E244 and K464, does not reproduce phenotypes similar to K464A or K464E, and, therefore, does not affect the function of K464.

**Pre-activated conformation of the CL-CNBD in K464A and K464E induces affinity change in CNBDs**. In HCN channels, a gating mechanism is proposed, in which for channel opening the CL-CNBD has to perform a leftward rotation relative to the channel pore to unwind the right-handed S6 helix bundle gate[6,18].

cAMP binding causes a leftward rotation of the CL-CNBD (Fig. 3b, c). Without additional energy supplied by voltage, however, the cAMP-triggered rotation is not sufficient to open the channel; yet, it is supportive for the voltage-induced rotation[6]. In this sense, cAMP causes the CL-CNBD to adopt a pre-activated or pre-disinhibited conformation.

In former studies employing patch-clamp fluorometry experiments, we and others showed that voltage-induced activation leads to increased cAMP binding affinity[27,28]. This increase of binding affinity preceded gate opening. Thus, it is not the actual gate opening that causes the affinity increase but preceding conformational changes[27]. However, the previous data could not be interpreted by underlying conformational changes.

The K464 mutants presented here provide a unique tool to probe if the leftward rotation of the CL-CNBD is the causative conformational change for the affinity increase. To this end, we measured the binding affinity of a fluorescently tagged cAMP derivative, 8-Cy3B-AHT-cAMP ($f_1$cAMP), in K464E with confocal patch-clamp fluorometry[29]. The results are summarized in Fig. 5.

The representative confocal images in Fig. 5a show patch pipettes carrying an excised macropatch expressing either wild-type mHCN2 or K464E channels at a non-activating and an activating voltage of −30 mV and −130 mV, respectively. The green fluorescence signal of the patch is caused by binding of 0.75 μM $f_1$cAMP to the channels. The red signal in the background is caused by the reference dye Dy647 (5 μM), used to subtract the background intensity of unbound 8-Cy3B-AHT-cAMP. Details of the subtraction procedure are described in the "Methods" section and in ref. Biskup et al.[30].

To monitor the affinity increase in response to channel activation, we applied, in analogy to the patch-clamp only experiments, a voltage jump from −30 to −130 mV, followed by a short test pulse of −100 mV and a deactivating pulse back to the holding potential of −30 mV. For each patch, the mean fluorescence intensity caused by a subsaturating $f_1$cAMP concentration was quantified for the dome of the patch and normalized to the individual maximum intensity caused by a saturating concentration of 2.5 μM, yielding $F/F_{max}$. Representative intensity-time courses of $F/F_{max}$, showing the ligand binding, and simultaneously measured current time courses, showing channel activation, are illustrated for K464E and mHCN2 in Fig. 5b. There was no binding increase for K464E upon channel activation.

To quantify this in more detail, we plotted $F/F_{max}$ against the $f_1$cAMP concentration to yield concentration-binding relationships for −30 and −130 mV (Fig. 5c). The Hill equation (Eq. (3)) was fitted to the mean data yielding the concentration of half-maximum binding, $BC_{50}$, of 0.40 μM for non-activated mHCN2 at −30 mV, 0.24 μM for activated mHCN2 at −130 mV, 0.28 μM for non-activated K464E at −30 mV, and 0.26 μM for activated K464E at −130 mV. The Hill coefficients, $H$, were 1.5, 1.6, 1.8, and 1.8, respectively. As shown previously, in mHCN2, the affinity in hyperpolarized channels at −130 mV was higher than for non-activated channels at −30 mV[27,28]. In K464E, there was no difference in cAMP affinity between activated and non-activated channels. Both values were similar to the affinity of activated mHCN2 wild-type channels.

To explain this, we determined changes in structural fluctuations within the CL-CNBD upon cAMP binding from our MD simulations and compared these changes with those induced by the K464E substitution. To this end, we computed the root mean square fluctuation (RMSF) of the side chains, a measure for atomic mobility, including all residues of the CL-CNBD. The side-chain mobility is expressed relative to the *apo* wild-type channel ($\Delta$RMSF; Eq. (4)). cAMP binding and K464E substitution lead to an overall rigidification of the CL-CNBD compared to

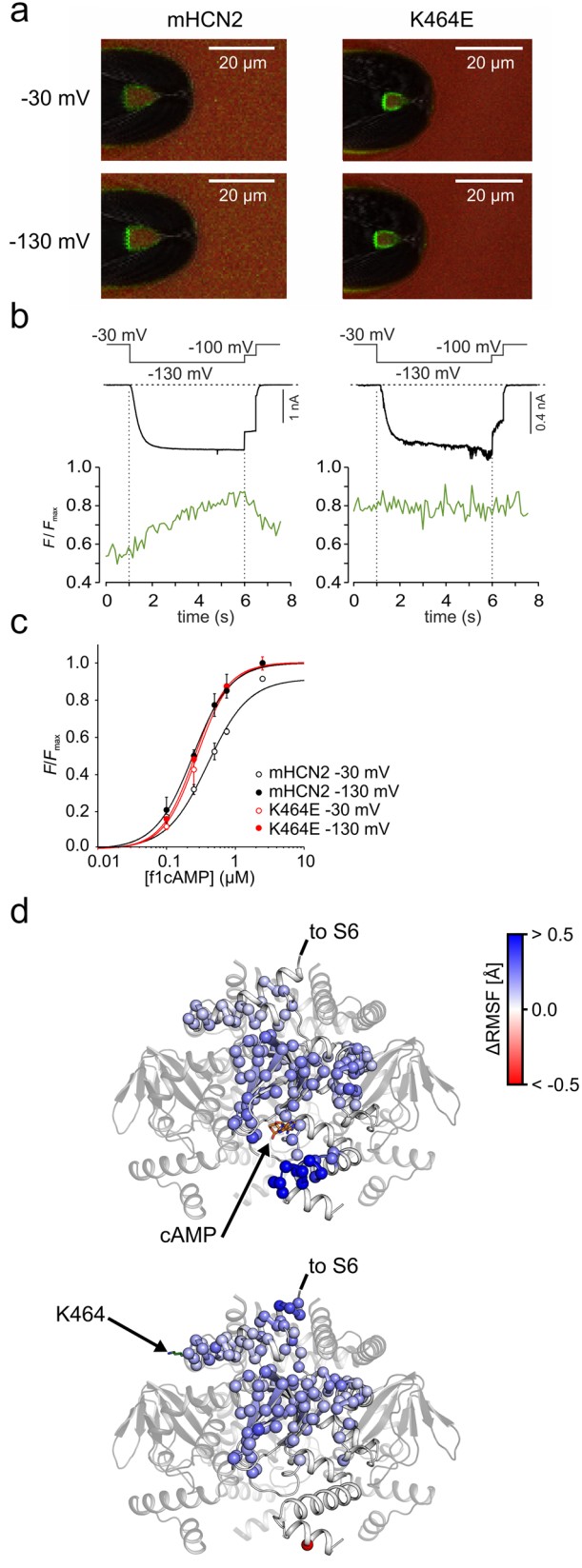

**Fig. 5 Activation-dependent affinities in K464 mutant and mHCN2 wild-type channels. a** Representative confocal images for mHCN2 and K464E. Upper panels show recordings at a non-activating voltage of −30 mV, lower panels at activating voltage of −130 mV. The tip of a patch pipette carrying a membrane patch expressing either mHCN2 or K464E is shown. The green signal is caused by 0.5 μM 8-Cy3B-AHT-cAMP binding to the channels, the red signal staining the background is caused by 5 μM Dy647, a reference dye required for subtracting the background signal of unbound 8-Cy3B-AHT-cAMP. **b** Time courses of simultaneously measured fluorescence increase (green) and current increase (black) following an activating voltage pulse from −30 to −130 mV. **c** Concentration-binding relationship for mHCN2 and K464E. The Hill function (Eq. (3)) is approximated to the averaged data ($n = 3$ to 6) yielding the concentration of half-maximum binding, $BC_{50}$, and the Hill coefficient, $H$. Error bars indicate SEM. **d** Isolated CL-CNBD shown as cartoon. Residues that behave significantly different to *apo* wild-type HCN2 are shown as spheres ($C_\alpha$ atoms only) and colored according to the residue-wise average ΔRMSF (see color bar on the right; see also Eq. (4); $n = 80$ independent replicas). Residues colored in blue are significantly more mobile in the *apo* wild-type channel ($p < 0.05$; $p$ value by $t$-test). Residues colored in red are significantly more mobile in the cAMP-bound wild-type channel (top panel) or *apo* K464E channel (lower panel) ($p < 0.05$; $p$ value by $t$-test). cAMP and K464 are shown as sticks.

of helix C in the CNBD is significantly less mobile, which may also explain why this region is not resolved in the hHCN1 in the absence of cAMP[6]. Thus, the data provide evidence that cAMP-binding to the CNBD is structurally and functionally coupled to the CL region, which, in turn, is in the direct vicinity of the voltage sensor. Inversely, the effect of CL modulation due to the K464E substitution is structurally and functionally connected to the CNBD. These changes in structural fluctuations are overlaid by a CL-CNBD rotation induced similarly by either the K464E substitution or cAMP binding (see above). The cumulative changes in structural dynamics of the CL-CNBD in the case of K464E thus generate a CL-CNBD state to which cAMP binding is more favorable.

From these data, we conclude that the leftward rotation of the CL-CNBD, caused either by the movement of the voltage sensors or by weakening interactions between the elbow region of the CL-CNBD and the HCN domain of the opposite subunit, is causative for the high-affinity state of the cAMP binding sites.

**K464 is indirectly involved in gate modulation.** It has been hypothesized that the rotation of the CL-CNBD could result in a displacement or unwrapping of the right-handed bundle of S6 helices, which harbors the gate-forming amino acids[6]. As we recorded a rotation of the CL-CNBD induced by K464E, which is similar to the rotation caused by cAMP binding[6,18], we now analyzed our MD simulations towards changes of structural features of the gate region associated with the rotation. In the mHCN2 channel, the gate is formed by I432, T436, and Q440 (Fig. 6a), which is identical to the architecture in hHCN4[18] (I510, T514, and Q518) (Fig. S7a). In hHCN1[6], by contrast, the iso-leucine is substituted by valine, such that the gate is formed by V390, T394, and Q398 (Fig. S7a). Because of the high structural similarity of isoleucine and valine, a comparison of the results across all three channels should still be appropriate.

To investigate whether the rotation of the CL-CNBD portion is associated with the gate opening, we measured the distances between two opposite gate residues of *apo* and cAMP-bound wild-type mHCN2, and K464E. One might assume that the CL-CNBD rotation induced by cAMP binding is associated with

the *apo* wild-type channel (Fig. 5d). Mapping ΔRMSF onto the structure of the CL-CNBD reveals that, in particular, the CL region and the β-roll of the CNBD, encompassing parts of the cAMP binding site, become less mobile in both cases. The magnitudes of ΔRMSF are in excellent agreement in both cases (Fig. S6). As a unique feature for cAMP binding, the C-terminus

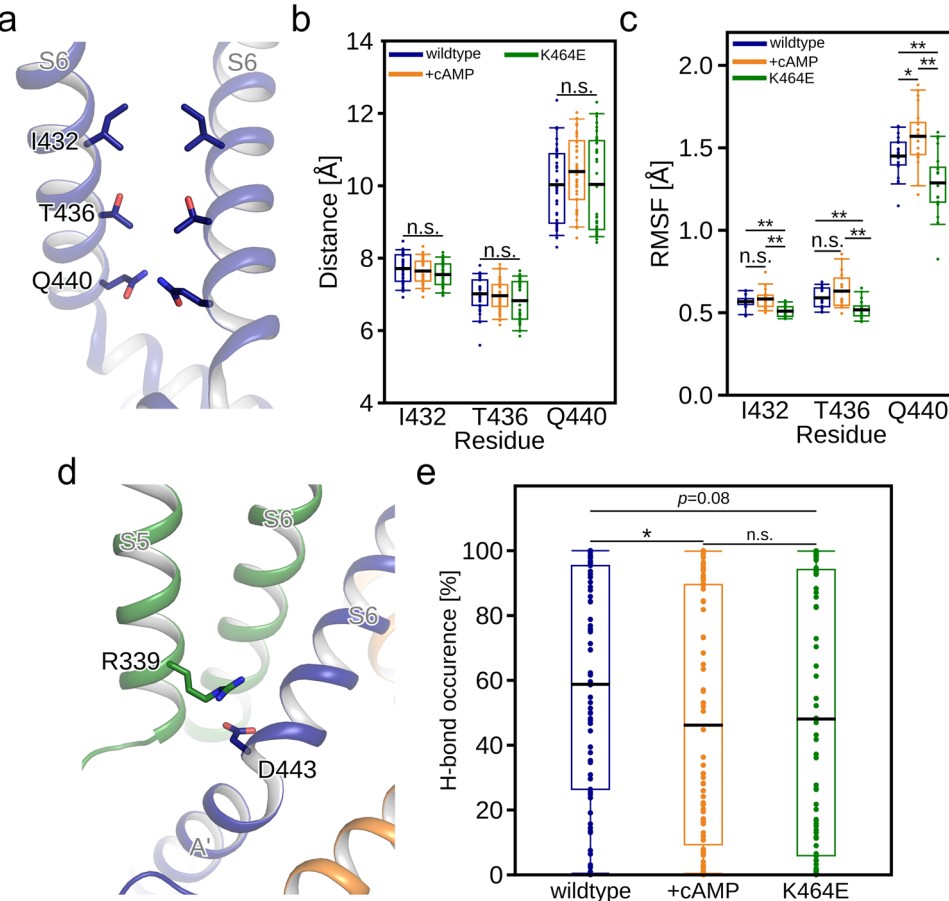

**Fig. 6 Conformational analyses of the S4/S5/S6 transmembrane portion of mHCN2. a** The gate region of the mHCN2 channel pore with only two opposite subunits is shown. **b** Distance measurements between the $C_\beta$-atoms of two opposite amino acids that form the gate. **c** Residue-wise root mean square fluctuations (RMSF) of gate-forming amino acids. All non-hydrogen side-chain atoms were considered. **d** The close-up view depicts the suggested interaction between D443 on subunit $i$ (blue) and R339 on $i + 1$ (green). **e** Average occurrence frequency of salt bridge interactions between residues depicted in panel (**d**). In **a** and **d**, relevant amino acids are shown as sticks. In **b**, **c**, and **e**, the individual data points are shown as circles and the mean values as black lines ($n = 40$ in **b** and $n = 80$ in **c** and **e**). The boxes denote the range from the 25th to 75th percentile and include 50% of the data points. The whiskers denote the 5th to 95th percentile and include 90% of the data points (*$p < 0.05$; **$p < 0.01$; n.s. not significantly different; $p$ value by $t$-test).

increased distances, although previous structural studies[6,18] suggest only marginal changes in pore diameter (Fig. S7b). Interestingly, also in the hyperpolarized conformation of hHCN1 the pore diameter remains almost unchanged, indicating a closed gate[31] (Fig. S7b). In line with this observation, during MD simulations, the pore diameter neither changes significantly upon cAMP binding nor after K464E substitution (Fig. 6b), suggesting a closed gate. Alternatively, CL-CNBD rotation and thus unwrapping of the S6 helical bundle might increase the mobility of the gate-forming side chains I432, T436, and Q440 in mHCN2, which would lower the resistance that cations face when passing the gate. To validate this, we calculated the RMSF considering all non-hydrogen side-chain atoms of the gate-forming amino acids. Upon cAMP binding, the side chain of Q440 at the intracellular terminus of S6 is significantly more mobile than the apo wild-type channel, whereas the opposite is seen for K464E (Fig. 6c). Similar changes are observed for I432 and T436, with all but the differences between apo and cAMP-bound wild-type channel being significant. Still, all differences are <0.2 Å. We, thus, conclude that structural changes of the CL-CNBD due to cAMP binding or the K464E substitution observed in our MD simulations have no relevant impact on the gate conformation in HCN channels, similar to what is observed in comparative analyses of experimental HCN structures.

Previous structural and mutational studies indicated a complex interplay between helices S4, S5, and S6 of the transmembrane portion upon hyperpolarization and gate opening[31–34] (Fig. S8a). Not surprisingly, in the absence of a hyperpolarizing voltage during MD simulations, the relative arrangement of helices S4 and S5 mimics that found under depolarized conditions in experimental structures (Fig. S8b, c). As to the S5 and S6 helices, salt bridges between these were found to be essential for stabilizing the closed channel, and alanine substitution of these residues favored channel opening[32,34]. In mHCN2, the salt bridge between R339 and D443 is partially lost upon cAMP binding to mHCN2 ($58.78\% \pm 3.99\%$ in the apo wild-type channel versus $46.19\% \pm 4.33\%$ in the cAMP-bound wild-type channel; $p < 0.05$), which is also the case in the K464E channel ($48.08\% \pm 4.54\%$; $p = 0.08$ relative to the apo wild-type channel) (Fig. 6d, e). The cAMP-bound wild-type channel does not behave differently from K464E (Fig. 6e). The reduced interaction frequency between D443 and R339 may explain how cAMP binding or K464E destabilize the closed gate.

## Discussion

In this study, we addressed the question of whether a direct interaction of opposite subunits is relevant for channel gating. To

this end, we combined mutagenesis, patch-clamp, confocal patch-clamp fluorometry, and MD simulations. Previous studies proposed a gating mechanism, in which for channel opening the CL-CNBD performs a leftward rotation to unwind the right-handed S6 helix bundle gate[6]. Such a leftward rotation is suggested to be induced by the movement of the voltage sensor upon a hyperpolarizing voltage jump. cAMP binding can cause a CL-CNBD leftward rotation, too. However, this cAMP-triggered rotation is not sufficient to open the channel but is supportive for voltage-induced rotation[6]. In this sense, cAMP causes the CL-CNBD to adopt a "pre-activated" or "less-inhibited" conformation.

Over the past years, a plethora of different residues has been identified that are involved in channel gating by mediating intra- and inter-subunit contacts. So far, only little is known about functionally relevant inter-subunit contacts between opposite subunits in HCN channels[15]. Close inspection of the recently published hHCN1 channel structure[6] or a structural model of mHCN2 (Fig. 1a) revealed the possibility that also opposite subunits might interact: The so-called elbow structure, a helix-turn-helix motif formed by the α-helices A′ and B′ of the CL[14], is near the HCN domain and the proposed position of the S2-S3 linker of the opposite subunit, which was not resolved in the 3D structure, suggesting pronounced flexibility for this region.

This elbow region is of particular interest as previous studies already suggested an essential role in channel activation via coupling the conformational changes in the CNBD to the channel pore[10,19,35–37]. It has been suggested that channel opening is related to a rotation of the CL around the central channel axis, due to the disruption of stabilizing interactions[20,21,38]. Recently it was suggested that the N-terminal HCN domain plays a role in channel activation upon hyperpolarization and cyclic nucleotide binding[15,39]. However, the purpose of the HCN domain-CL interaction on channel activation is not entirely understood.

At the very tip of the elbow structure, we identified a lysine residue in mHCN2, K464, which is a key player in controlling the CL-CNBD rotation. To study this residue in detail, we either neutralized or reversed the positive charge by constructing the mutants K464A and K464E. In K464E, steady-state parameters suggested that in the absence of cAMP the closed state was destabilized, such that the behavior of the empty K464E resembled the behavior of a cAMP-saturated wild-type channel. In K464A, such a destabilization was visible to a lower degree, suggesting that interactions between negative charges foster closed-state destabilization. Interestingly, while the effect of cAMP on the steady-state parameters was strongly reduced in K464E, the activation kinetics was still cAMP-dependent, similarly to what was found for mHCN2.

One possible explanation of the electrophysiological experiments is that the CL-CNBD in K464E adopts a conformation similar to a cAMP-bound wild-type channel, possibly by being pushed into a leftward rotation due to repulsive forces from the opposite subunit. To corroborate this hypothesis, we performed unbiased MD simulations of *apo* wild-type mHCN2, wild-type mHCN2 bound to four cAMP molecules, and the *apo* K464E variant of the mHCN2 channel. The simulations revealed that the side chain of K464 forms a hydrogen bond with the backbone of M155 in the HCN domain, thereby bridging two opposite subunits. This hydrogen bond is significantly weakened upon K464 substitution in K464E. Furthermore, going from *apo* mHCN2 to mHCN2 bound to four cAMP molecules to the K464E variant, an increasing rotation angle of the CL-CNBD in a leftward direction was found, which agrees in terms of direction and magnitude with structural changes induced by cAMP binding to homologous HCN channels[6]. One might argue that the small differences between the apo K464E mHCN2 simulations may be due to inherent limitations of the unbiased MD simulation.

However, we followed the extensively validated "ensemble average approach", a procedure to interpret equilibrium dynamics from multiple, independent MD trajectories[40–42], to minimize the potential impact of insufficient sampling. Moreover, we started the MD simulations from the same channel conformation, allowing to interpret results between the mHCN2 simulations on a relative basis. The observation that the activation kinetics was still cAMP-dependent suggests that the conformational changes, which are rate-limiting for channel activation, are different from the described rotational movement of the CL-CNBD portion.

An interaction between the backbone of M155 and the side chain of K464 was also suggested by Porro and coworkers (2019). They studied the mutant K464A, but not K464E, and observed a similar shift of $V_{1/2}$ under cAMP-free conditions as presented herein. They found that the cAMP effect is only lost after simultaneously breaking a second bond (E478-R154), which connects the C-linker with the HCN-domain of the adjacent subunit. In our simulations, the salt bridge between E478-R154 is insensitive to cAMP binding but significantly weakened in the K464E channel (Fig. S9). However, the interaction frequencies are >70% throughout all investigated channels and simulations, suggesting that this interaction may be relevant for stabilizing adjacent subunits rather than being essential for cAMP-induced channel disinhibition. This assumption is supported by the functional data of Porro et al. on E478A HCN2, which is still highly sensitive to cAMP binding and almost identical to the wild-type channel[15]. How cAMP influences the R154A channel was not tested. Interestingly, the functional data on channel activation also suggests that the K464A-E478A resembles the *apo* wild-type channel[15], while our K464E resembles the cAMP-bound wild-type channel (Fig. 2). Our MD simulations suggest that the K464E channel adopts a conformation similar to the cAMP bound wild-type channel. One might speculate that K464A-E478A HCN2 adopts a conformation similar to the *apo* wild-type channel, which provides a plausible but not yet proven structural explanation for the functional data.

The experimental structures of hHCN1[6] and our MD simulations on mHCN2 revealed the backbone of M155 in the second α-helix of the opposite HCN domain as an interaction partner. To exclude that the side-chain of M155 is involved in interactions with K464, we mutated the methionine to alanine, arginine, or glutamate and showed the effects of these substitutions on steady-state activation. The effects were only mild in all cases and, importantly, did not mimic the phenotype of K464E. We thus concluded that the side-chain of M155 is not primarily involved in interactions with K464.

We next studied two negatively charged residues in the opposite S2/S3 linker, D244 and E247, which herein had also been identified as potential interaction partners for K464. However, neither neutralization nor charge reversal at those positions resulted in phenotypes similar to K464A or K464E. We thus concluded that the function of K464 in mHCN2 wild type does not rely on the formation of a bond with either one of those residues. To further support this conclusion for the residue E247, which showed the highest likelihood of developing a salt bridge with K464, we confirmed with the mutant E247R_K464E that rescuing a possible salt bridge does not result in a wild-type-like phenotype. This construct still behaved like the K464E mutant, possibly because the proposed repulsive forces introduced by the glutamate at position K464 find a counterpart in other negatively charged residues near position 247, for instance, D244. This finding may suggest that D244 or E247 take over the role of an interaction partner of K464 if the respective other residue is substituted (as in the case of the D244 and E247 mutants).

Notably, all four constructs showed a shift of $V_{1/2}$ to more negative values in the absence of cAMP, indicating a stabilization

of the closed state. Because $V_{1/2}$ at saturating cAMP is less affected, for all constructs, the cAMP-induced $\Delta V_{1/2}$ is more pronounced than for mHCN2 wild type. Thus, the negatively charged residues in the S2-S3 linker seem to be involved in voltage-dependent rather than in cAMP-dependent gating.

All our results suggest that in the K464E channel the CL-CNBD is rotated in a leftward direction compared to an *apo* wild-type channel. With this tool in our hands, we studied if this leftward rotation is crucial for inducing the activity-dependent increase of cAMP affinity[27,28]. To this end, we performed cPCF experiments in K464E employing a fluorescently tagged cAMP derivative, 8-AHT-Cy3B-cAMP[29]. Interestingly, in K464E, there was no difference in cAMP affinity between activated and non-activated channels. Both cAMP affinities were similar to the cAMP affinity of activated mHCN2 wild-type channels. We concluded that the leftward rotation of the CL-CNBD, caused by weakening interactions between the elbow region of the CL-CNBD and the HCN domain of the opposite subunit, is causative for the high-affinity state of the cAMP binding sites.

As we identified a rotation of the CL-CNBD portion induced by K464E similar to the rotation caused by cAMP binding[6,18], we used K464E as a tool to analyze the effect of this rotation on the inner S6 helix bundle gate. In our experiments, channel activation is favored upon cAMP binding or after K to E substitution. One might assume that this is related to a widening of the gate, which, in turn, facilitates cation passage of the gate more efficiently.

Indeed, by using the full-length hHCN1 structure, Gross and coworkers proposed a model in which, upon cAMP binding, the elbow moves to the shoulder of the adjacent subunit in an overall centrifugal motion away from the central axis of the channel pore, which might lead to a widening of the channel pore[37]. However, the authors also stated that with their approach, they could not gauge whether the widening is sufficient to open the intracellular channel gate. Interestingly, our simulation data showed no direct modulation of the gate conformation, neither by K464E substitution nor upon cAMP binding. Thus, our simulation data mirrors the structural observations from high-resolution cryo-EM structures of the hHCN1[6] and hHCN4[18]. Both channels were structurally resolved in the presence and absence of cAMP, and in neither of the channels did cAMP binding lead to a widening of the inner channel gate, underscoring the excellent agreement of simulation and experimental data. The latter result is in contrast to previous studies, reporting that cAMP-induced rotation of the CL disk leads to a consequent widening of the inner channel gate[21,43]. These studies have been performed on truncated channels lacking the transmembrane portion, however. In the most recent cAMP-bound hHCN1 structure, where a hyperpolarized state was reached through chemical cross-linking, the gate also remains closed[31].

If there is no direct effect of cAMP-induced CL-CNBD rotation on the gate, what is the purpose of this rotation for HCN channel modulation? In HCN channels, the S6 helical bundle is in direct neighborhood to helices S5 and S4[6,18,31] (Fig. S8). On the time-scales of our MD simulations, we did not see a change in the spatial organization of these three helices, induced by neither cAMP nor K464E (Fig. S8b). Comparative analyses of hHCN1 structures in *apo* and depolarized, cAMP-bound and depolarized, and cAMP-bound and hyperpolarized conformations confirm this observation[6,31] (Fig. S8a). By contrast, the conformations of helices S4 and S5 are highly sensitive to hyperpolarization[6,31] (Fig. S8a). However, in the absence of a hyperpolarizing voltage in our MD simulations, the relative arrangement of helices S4 and S5 mimics that found under depolarized conditions in experimental structures in all investigated states of mHCN2 (Fig. S8b, c). A possible way for how cAMP binding- or K464E-induced CL-CNBD rotation can, at

least, indirectly destabilize the closed gate is by reducing stabilizing interactions between R339 on S5 and D443 on S6 (Fig. 6d, e). The inter-subunit salt bridge between D443 and R339 locks the closed channel conformation[32,34]. CL-CNBD rotation, induced by either cAMP binding or K464E substitution, breaks the salt bridge and, thus, removes or at least weakens the structural restraints of the channel gate, which, in turn, may facilitate channel opening under hyperpolarization conditions. Recently, MD simulations of a truncated HCN channel, in which only the TM portion of the channel was considered, in the presence of a hyperpolarizing voltage recorded the S4 helix movement[33]. Although not yet possible for full-length HCN channels, such simulations together with different rotation states of the CL-CNBD will pave the way to scrutinize the interplay of hyperpolarization-mediated activation and cyclic nucleotide-, or, as in the case of K464E, substitution-, mediated gating of HCN channels at the atomistic level.

In summary, opposite subunits in HCN channels are functionally coupled. Our data suggest that the interactions between opposite subunits, embedded in a network of interactions between adjacent subunits[15,16], are relevant for the autoinhibitory properties of the channels. Both cAMP binding and charge inversion of a single amino acid drive the channel into a conformation that weakens autoinhibition.

## Methods

**Xenopus laevis oocytes as heterologous expression system**. The surgical removal of oocytes was performed under anesthesia (0.3% tricaine methanesulfonate (MS-222) (Pharmaq Ltd., Fordingbridge, UK) from adult female South African claw frog *Xenopus laevis* (Nasco, Fort Atkinson, US). The oocytes were treated with collagenase A (3 mg/ml; Roche, Grenzach-Wyhlen, Germany) for 105 min in Ca²⁺-free Barth's solution containing (in mM) 82.5 NaCl, 2 KCl, 1 MgCl₂, and 5 Hepes, pH 7.5. Oocytes of stages IV and V were manually dissected and injected with cRNA encoding either mHCN2 channels of Mus musculus or the mHCN2 mutants K464E, K464A, M155A, M155R, M155E, E247A, E247R, E247R_K464E, D244A, D244K, respectively. After injection with cRNA, the oocytes were incubated at 18 °C for 2–6 days in Barth's solution containing (in mM) 84 NaCl, 1 KCl, 2.4 NaHCO₃, 0.82 MgSO₄, 0.41 CaCl₂, 0.33 Ca(NO₃)₂, 7.5 TRIS, pH 7.4. Oocytes harvested in our own lab were complemented with ready-to-use oocytes purchased from Ecocyte Bioscience (Dortmund, Germany). The surgery procedures were carried out in accordance with the German Animal Welfare Act with the approval of the Thuringian State Office for Consumer Protection on 30.08.2013 and 09.05.2018.

**Molecular biology**. The mouse *HCN2* (UniProt ID O88703 including two modifications, G8E and E55G without functional relevance) and all modified subunit variants were subcloned in front of the T7 promoter of pGEMHEnew. Point mutations K464E, K464A, M155A, M155R, M155E, E247A, E247R, E247R_K464E, D244A, and D244K were introduced via the overlapping PCR-strategy followed by subcloning of the modified fragment using flanking restriction sites. Correctness of the sequences was confirmed by restriction analysis and sequencing (Microsynth SEQLAB, Göttingen, Germany). cRNAs were prepared using the mMESSAGE mMACHINE T7 Kit (Ambion Inc, Austin, USA).

**Electrophysiological experiments**. Macroscopic currents were recorded using the patch-clamp technique in the inside-out configuration. All measurements were started after a delay of 3.5 min to minimize run-down phenomena. Patch pipettes were pulled from quartz tubings whose outer and inner diameters were 1.0 and 0.7 mm (VITROCOM, New Jersey, USA), respectively, using a laser puller (P-2000, Sutter Instrument, Novato, USA). The pipette resistance was 1.2–2.1 MOhm. The bath solution contained (in mM) 100 KCl, 10 EGTA, and 10 Hepes, pH 7.2, and the pipette solution contained (in mM) 120 KCl, 10 Hepes, and 1.0 CaCl₂, pH 7.2. For parts of the experiments, a saturating concentration of 10 μM cAMP (BIOLOG LSI GmbH & Co KG, Bremen, Germany) was applied with the bath solution. An HEKA EPC 10 USB amplifier (Harvard Apparatus, Holliston, USA) was used for current recording. Pulsing and data recording were controlled by the Patchmaster software (Harvard Apparatus, Holliston, USA). The sampling rate was 5 kHz. The holding potential was generally −30 mV. Each recording was performed in an individual membrane patch. Maximally two membrane patches were excised from one individual oocyte. For steady-state activation curves, relative current values for each recording were fitted individually (see the "Quantification and statistical analysis" section).

**Confocal patch-clamp fluorometry**. The binding of the fluorescently tagged cAMP derivative 8-Cy3B-AHT-cAMP (f$_1$cAMP)[29] and the ionic current in macropatches were measured simultaneously by confocal patch-clamp fluorometry (cPCF) as described previously[27,30]. The patch pipettes were pulled from borosilicate glass tubing with an outer and inner diameter of 2.0 and 1.0 mm, respectively (Hilgenberg GmbH, Malsfeld, Germany). The pipette resistance was 0.7–1.2 MΩ. The bath solution contained (in mM) 100 KCl, 10 EGTA, and 10 Hepes, pH 7.2, and the pipette solution contained (in mM) 120 KCl, 10 Hepes, and 1.0 CaCl$_2$, pH 7.2. Ionic currents were recorded using an Axopatch 200B amplifier (Axon Instruments, Foster City, USA). Current measurements were controlled and data were collected with the ISO3 software (MFK, Niedernhausen, Germany). The sampling rate was 2 kHz. Fluorescence imaging was performed with an LSM 710 confocal microscope (Zeiss, Jena, Germany) and was triggered by the ISO3 software (MFK, Niedernhausen, Germany). To subtract the fluorescence of the unbound f$_1$cAMP from that of the bound f$_1$cAMP, a second dye, DY647 (Dyomics, Jena, Germany), was added to the bath solution at a concentration of 5 μM. f$_1$cAMP and DY647 were excited at 543 nm and 633 nm, respectively, and detection bands of 546–635 nm and 637–759 nm were selected. Before subtraction, DY647 fluorescence intensity was scaled on f$_1$cAMP fluorescence intensity in the bath and the pipette interior. The fluorescence intensity in the patch dome only was used to quantify the portion of bound ligands[30]. A concentration of 2.5 μM f$_1$cAMP was used for saturating all four binding sites and thus, for quantifying the maximum fluorescence, $F_{max}$. For concentration-binding curves, each membrane patch was exposed to the saturating concentration and, additionally, to one, two, three, or four subsaturating concentrations. The resulting relative fluorescence intensities were averaged and the averaged values were fitted (see "Quantification and statistical analysis").

**Quantification and statistical analysis**. Steady-state activation curves were analyzed by fitting the Boltzmann equation to each individual recording using the OriginPro 9.0G software (Northampton, USA):

$$I/I_{max} = I/I_{max,satV}/[1 + \exp(z\delta F(V - V_{1/2})/RT)] \qquad (1)$$

$I/I_{max}$ is the relative current, $I/I_{max,satV}$ is the relative current at a saturating voltage, $V_{1/2}$ is the voltage of half-maximum activation, and $z\delta$ is the effective gating charge. $F$, $R$, and $T$ are the Faraday constant, the molar gas constant, and the temperature in Kelvin, respectively.

The time courses of current activation and deactivation were fitted with a single exponential starting after an initial delay using the OriginPro 9.0G software (Northampton, USA):

$$I(t) = A\exp[-t/\tau] \qquad (2)$$

$A$ is the amplitude, $t$ the time, and $\tau$ the time constant for activation and deactivation, respectively.

Concentration-binding relationships were analyzed by approximating the Hill equation to averaged data using the OriginPro 9.0G software (Northampton, USA).

$$F/F_{max} = 1/[1 + (BC_{50}/[\text{agonist}])^H] \qquad (3)$$

$F$ is the actual fluorescent intensity at a given f$_1$cAMP concentration, $F_{max}$ the maximal fluorescent intensity at a saturating concentration of 2.5 μM f$_1$cAMP, $BC_{50}$ the concentration of half-maximum binding, and $H$ the Hill coefficient of binding. Fluorescent intensities were generally obtained from the steady-state phase of a voltage pulse. Values for $BC_{50}$ and $H$ were yielded once for the averaged $F/F_{max}$ data.

Experimental data are given as mean ± standard error of mean (SEM). Statistical analysis was performed by an unpaired Student's $t$-test. A value of $p < 0.05$ was accepted as statistically significant.

**System setup for molecular dynamics simulations**. As currently no experimental 3D structure of the full-length mHCN2 is available, we carried out MD simulations using a homology model of mHCN2 (amino acid sequence taken from UNIPROT: O88703) based on the homologous and ligand-free hHCN1 structure[6] (PDB: 5U6O); for generating the homology model, Maestro® from the Schrödinger suite for molecular modeling (release 2018-3)[44] was used. The mHCN2 structure was built for the sequence from L136 to D650, and the sequence identity between mHCN2 and hHCN1 is 80%. As to the CNBD of the ligand-free hHCN1 structure[6], the structures of helices C and D were not resolved completely, such that these two helices were prepared on the basis of the ligand-bound hHCN1[6] and afterward connected to the rest of the mHCN2 structure. Thus, the final homology model of mHCN2 contains the HCN domain, the transmembrane helices, the CL, and the CNBD up until helix D.

The full-length mHCN2 was further prepared for simulations by using the Protein Preparation Wizard[45] distributed with Maestro® (release 2018-3)[44]. First, we added ACE and NME groups to the channel's termini to avoid artificially charged termini. Protonation states were assigned according to the physiological pH of 7.4 and pKa values computed by PROPKA[46,47]. That way, all glutamate and aspartate residues are deprotonated and negatively charged, all lysine and arginine residues are protonated and positively charged. As to histidine residues, H474 was assigned to the HIP state (net charge +1 with hydrogen atoms bound at both imidazole nitrogen atoms), H178, H328, and H397 to the HIE state (net charge ±0

with a hydrogen atom bound at the ε-nitrogen atom), and the remaining histidine residues to the HID state (net charge ±0 with a hydrogen atom bound at the δ-nitrogen atom). All hydrogen atoms were added according to the Amber ff14SB library[48]. Finally, the prepared mHCN2 structure was assembled to a homotetrameric channel, again by using the 3D structure of the ligand-free hHCN1[6] as the template. The homotetrameric mHCN2 channel was inserted into a DOPC bilayer by using PACKMOL-Memgen[49]. After adding 0.4 mM KCl, the systems were solvated with SPC/E water[50], also by using PACKMOL-Memgen[49] (this structure is further referred to as the wild-type mHCN2 channel). Initial coordinates for cAMP molecules were adapted from cAMP-bound structure of the hHCN1 channel. Therefore, we first aligned the CNBD backbone of the cAMP-bound hHCN1 structure to the ligand-free hHCN1 structure and translated the cAMP coordinates accordingly. As the differences between the two structures are small, no structural clashes between any cAMP atom and any channel atom was observed. To investigate also the influence of the K464E mutation on mHCN2 channel activation and deactivation, we also prepared a mHCN2 structure carrying the K464E mutation (this system is further referred to as the K464E mHCN2 channel). Finally, to investigate whether homology modeling of the mHCN2 introduces structural artifacts, we also prepared the *apo* wild type, cAMP-bound wild type, and *apo* K422E variant of the hHCN1 channel, which served as the template for homology modeling.

**Molecular dynamics simulations**. MD simulations were performed using Amber18[51]. The Amber ff14SB force field[48] was used for the mHCN2 and hHCN1 channels, lipid17 for the DOPC lipids, parameters from Joung and Cheatham[52] for K$^+$ and Cl$^-$, and SPC/E[50] for the water molecules.

The detailed minimization, thermalization, and equilibration protocol is reported in ref. Kater et al.[53]. In short, all structures were initially subjected to three rounds of energy minimization to resolve steric clashes. The system was then heated to 300 K, and the pressure was adapted to 1 bar. During thermalization and pressure adaptation, we kept the protein atoms fixed by positional restraints of 1 kcal mol$^{-1}$ Å$^{-2}$, which were gradually removed. Finally, an NPT simulation at 300 K and 1 bar of 50 ns length was performed with the unrestrained systems. Using the resulting structures as starting points, we performed 20 independent NPT production simulations at 300 K and 1 bar for 1 μs each. The initial velocities were randomly assigned during the first step of the NPT production simulation, such that each simulation can be considered as an independent replica. During production simulations, Newton's equations of motion were integrated in 4 fs intervals, applying the hydrogen mass repartitioning approach[54] to all non-water molecules. Water molecules, by contrast, were handled by the SHAKE algorithm[55]. Coordinates were stored into a trajectory file every 200 ps. The equilibration simulations were performed using the pmemd.MPI[56] module from Amber18[51], while the production simulations were performed with the pmemd.CUDA module[57].

**Trajectory analysis**. All trajectories were analyzed with cpptraj[58] from Amber-Tools18. To investigate the interactions between two opposite subunits, we measured hydrogen bond interactions between K464 from one subunit to M155, E243, D244, and E247 from the other subunit. Hydrogen bond interactions were determined using a distance of 3.5 Å between the two donor and acceptor atoms and an angle (donor atom, H, acceptor atom) of 120° as cutoff criteria[59].

To investigate the rotation of the CL-CNBD relative to the channel pore, but also relative to the starting structure, we measured the dihedral angle as indicated in Fig. S10. Technically, the four reference points to define a dihedral angle were defined as follows: (I) the center of mass (COM) of C$_\alpha$-atoms of the four terminal residues (M460-H463) of the A′-helix of the CL, (II) the COM of C$_\alpha$-atoms of T436 on the S6 helix of each subunit, (III) the COM of C$_\alpha$-atoms of I432 on the S6 helix of each subunit, and (IV) again the COM of C$_\alpha$-atoms of the four terminal residues (M460-H463) of the A′-helix of the CL. The reference points I–III are static and were defined on the basis of the starting structure, while the reference point IV was determined over the course of the trajectory. That way, after superimposing the channel pore, a movement of IV will change the dihedral angle and can be recorded as a rotation of the CL-CNBD. Finally, we also calculated the average structures over the full ensemble of *apo* and cAMP-bound wild-type mHCN2 and the K464E variant.

To investigate changes in side-chain mobility, we calculated the residue-wise root mean square fluctuation (RMSF), including all non-hydrogen side-chain atoms. To determine the changes relative to *apo* wild-type HCN2, we calculated ΔRMSF according to Eq. (4)

$$\Delta RMSF = RMSF_{apo,wildtype} - RMSF_{\{cAMP,wildtype|apoK464E\}}, \qquad (4)$$

where $RMSF_{apo,wildtype}$ is the mean residue-wise RMSF of the *apo* wild-type channel and $RMSF_{\{cAMP,wildtype|apo\ K464E\}}$ is the mean residue-wise RMSF of the cAMP-bound wild-type channel or the *apo* K464E channel. If $RMSF_{\{cAMP,wildtype|apo\ K464E\}}$ is not significantly different (in the case of $p > 0.05$; $p$ value by $t$-test) to $RMSF_{apo,wildtype}$, ΔRMSF was reset to zero, which allows focusing on the significant changes only.

To investigate gate modulation, we measured the distance between C$_\beta$-atoms of two opposite amino acids that form the gate. In addition, we measured the RMSF

of all non-hydrogen side-chain atoms of amino acids that form the gate. In mHCN2 the gate is formed by I432, T436, and Q440.

Hydrogen bond interactions, RMSF values, distances, and rotation angles are reported as mean value ± SEM. As the mHCN2 channel is a homotetramer and we performed 20 independent MD runs, the mean values and SEMs were calculated for $n = 80$ independent measurements, if not stated differently. A two-sample $t$-test between mean values of *apo* and cAMP-bound wild-type mHCN2 and the K464E variant was performed, and *p*-values < 0.05 were considered significant.

**Statistics and reproducibility**. Statistical comparison of mutated channels and mHCN2 wild-type channels regarding their electrophysiogal parameters were performed with the two-tailed unpaired Student $t$ test, using the analyis software OriginPro 9.0G software (Northampton, USA). In addition, we used NumPy[60] for comparison of the MD simulation data. Exact numbers of measurements $n$ included are always provided in the respective figure legends. *p*-values equal or inferior to 0.05 were considered statistically significant.

**Reporting summary**. Further information on research design is available in the Nature Research Reporting Summary linked to this article.

## Data availability

The datasets generated during and/or analyzed during the current study are available under the following link: https://doi.org/10.25838/d5p-27.

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

## Acknowledgements

We thank Sandra Bernhardt, Uta Enke, Andrea Kolchmeier, Claudia Ranke and Karin Schoknecht for excellent technical assistance. This work was supported by the DFG Research Unit 2518 DynIon of the Deutsche Forschungsgemeinschaft (projects P2 (KU 3092/2-1, BE 1250/19-1) and P7 (GO 1367/2-1)). We are grateful for computational support and infrastructure provided by the "Zentrum für Informations- und Medientechnologie" (ZIM) at the Heinrich Heine University Düsseldorf and the computing time provided by the John von Neumann Institute for Computing (NIC) to H.G. and B.F. on the supercomputer JUWELS at Jülich Supercomputing Centre (JSC) (user IDs: HKF7; HDD17).

## Author contributions

M.K. carried out electrophysiological and cPCF experiments and analyzed data. B.F. carried out the molecular simulation studies, analyzed data, prepared figures, and wrote the manuscript. S.Y. carried out electrophysiological and cPCF experiments. T.S. and C.S. engineered the mutants. M.L and A.S. synthesized fluorescent cyclic nucleotides. R.S. designed experiments. K.B. designed experiments. H.G. designed molecular simulation studies, analyzed data, and wrote the manuscript. J.K. carried out electrophysiological and cPCF experiments and analyzed data, designed experiments, prepared figures, and wrote the manuscript.

## Funding

## Competing interests

The authors declare no competing interests.
