## [Peer Review File · Communications Biology]

Reviewers' comments:

Reviewer #1 (Remarks to the Author):

Report on manuscript: Functional and structural characterization of interactions between opposit subunits in HCN pacemaker channels by Kondapuram and coworkers

The present study combines functional data, mutation analysis and computational studies for understanding the challenging question on how allosteric gating in HCN channels works. A pertinent question in the context of this gating mechanism is how the information of conformational changes are transduced from the ligand binding site into the channel part, which hosts the voltage sensitive gates. By scrutiny of the cryEM structure of the HCN1 channel the authors identify a conserved cationic amino acid in a critical position of the cytosolic domain, which might play a crucial role in this mechanical coupling. The experimental and computational studies indeed underpin a functional role of this cationic amino acid (K464). The data suggest that the Arg forms a hydrogen bond with the backbone of Met155 and that this coupling affects voltage and cAMP dependent gating.

As a side effect of the MD simulations the authors realized that K464 can also form salt bridges with anionic AA in the S2/S3 linker of the channel. In a mutant channel K464E this salt bridge formation is abolished with the effect that the CNBD rotates around the central axis of the channel. Since a similar rotation is seen between HCN structures with/without cAMP the authors speculate that this interaction could be an additional component in the mechanic transmission between ligand binding domain and channel gating.

The data are generally interesting and they show with no doubt functional of K464 in gating of HCN2. A positive aspect of the manuscript is the attempt to correlate functional/mutational data with potential conformational changes in the channel protein. But a critical evaluation of the data make me conclude that the data are heavily overinterpreted. In the context of published data (see below) I don't see how the entire mechanism of HCN2 gating can be interpreted in the context of a single amino acid. Also, while I find the computational data interesting they are too weak (see below) to make these strong points. For this reason I have to reject publication of the manuscript in the present form.

Major concerns:

- 1) The authors use MD simulations of a homology model based on the experimental HCN1 structure to examine atomic interactions of K464 within the channel protein. Since homology models bare the hazard of introducing structural artifacts and hence false positive interactions in a protein they should have performed a control simulation with the HCN1 structure. If as the authors imply the structural motive of interaction is universal in HCN channels they should have found the same interactions in the real structure. This would greatly support the impact of the computational data from the homology model.
- 2) The authors show that the occupation time of the hydrogen bond between K464 and its partners M155 or E247 is less than 25 %. This means, an interaction which is considered so crucial for structure/function relationship of HCN channels is much more absent than present. Notably in studies of other proteins the occupation time of critical salt bridges and/or hydrogen bonds is often in the order of 60 to 100%.
- 3) Furthermore, on the same issue, I do not understand whether this H+ bond frequency was calculated individually for each subunit or as a cumulative measure for all 4 subunits. In latter case the 25% frequency would have to be divided by 4. An interaction frequency of 5% for each subunit would not be more than a random event.
- 4) The data in Fig. 3B are interpreted as evidence for a very small rotation or small rotation of the CNBD in presence of cAMP or in the K/E mutant respectively. I understand that this information is derived from the relative distance of the peaks of the distribution. If these narrow distributions with a small width I might understand this argument. But the distributions are very broad, meaning that even in the mutant the orientation of the CNBD is not distinguishable from the wt.
- 5) The data in Fig. 3 imply that the K464 is able to interact with M155 and E247. If one of the interactions is eliminated by mutation of E247mutation one would expect that the frequency of interacting with M155 is favored. Is this observed in the MD simulations and reflected in the functional data?
- 6) The authors present their work as if they were the first to discover the importance of AA K464 and a functional interaction between the elbow and the CNBD in HCN channels. This is not the case. A simple search with the key word "K464 and HCN2" in Google shows that this AA was

already mentioned by Pater et al. as a putative site for sumoylation (10.3389/fnmol.2016.00168). This is probably not important for the present analysis but it should be mentioned for completeness. More important however is the fact that Porro and coworkers (DOI: 10.7554/eLife.49672) already mention an interaction between M113-K422 in HCN1 (which is equivalent to M155 and K464 in HCN2). In their study individual and double mutations of K464 and E478 residues to alanines generated opposite effects on the voltage dependence of the channel. These data imply that the situation is not as straight forward as presented here. Not the same amino acids but a similar concept namely the interaction between C-linker and CNBD was also examined by Wang et al (doi: 10.1074/jbc.RA120.013281). In this study it was found that three residues on the C-linker/CNBD (E478, Q382 and H559) make direct interactions with residues R154 and S158 on the HCND. Disrupting these interactions affected both voltage- and cAMP-dependent gating of HCN2 channels. The critical AA are so close to the key residue of this study, K464, that these results cannot be ignored in the discussion of the present paper.

Reviewer #2 (Remarks to the Author):

In this study, based on the analysis of the HCN1 channel structure and homology model of HCN2 channels, authors identified interactions between K464 on the C-linker of one subunit and M155 on the opposite subunit in HCN2 channels. K464E mutation shifted the current versus voltage plots to more depolarized potentials and decreased the cAMP sensitivity of the channel. MD simulations further supported direct interactions between M155 and K464 residues and suggested that K464E mutation introduces a rotation of the CL-CNBD in the same direction as induced by cAMP. Analysis of a cAMP fluorescent analog binding to the WT and K464E channels using patch clamp fluorometry indicated that the affinity of cAMP binding was the same for the activated (-130 mV) and non-activated (-30 mV) K464E mutant channels, and for the activated WT channels. Importantly, this binding affinity was almost two-fold higher than the cAMP affinity for WT non-activated channels. This suggests that the rotation of the CL-CNBD induced by K464E mutation increases cAMP binding affinity, supporting a hypothesis that voltage induced activation of HCN channels, also associated with similar rotation, increases cAMP binding affinity.

This is a very elegant study that for the first time focuses on examination of interactions between residues on the opposite subunits in HCN channels, while previous studies were focused on the interactions between residues on the adjacent subunits. The finding that the K464/M155 interaction favors the closed state of the channel, and destabilizing this interaction with K464E mutation favors an open state of the channel and mimics cAMP effect is interesting and important for understanding the gating mechanism of HCN channels. The data are high quality and findings are substantiated with complementary methods. However, there are some issues that need to be addressed.

Major comments:

1. Please include current traces for the mutant channels K464E and K464A in the absence and presence of cAMP.
2. Although K464E mutation decreased the differences in the current versus voltage (I/V) plots and activation kinetics in the absence and presence of cAMP, and brought these values closer to the ones for WT HCN channels in the presence of cAMP, Figs. 2B and 2C show noticeable differences. Therefore, phrases "largely superimposed" and "largely similar" should be avoided, unless there is a rigorous statistical analysis done to indicate that the data in the absence and presence of cAMP are not statistically different.
3. Although the authors focus on the interactions between M113 on the HCN domain of one subunit and K464 on the C-linker of the opposing subunits, Wang et al. (JBC, 2020) investigated interactions between R112 on the HCN domain of one subunit and E478/Q478 on the C-linker of the adjacent subunit, and Porro et al (2019) considered interactions between R112 and E478 from the adjacent subunit. R112 and M113 are adjacent residues and the global conformational changes happening during channel activation are bound to affect interactions between both R112 and

M113. The proximity of the interactions between the opposite and adjacent subunits indicates the strategic location of these residues important for the channel gating. These previous findings should be discussed in the manuscript.

4. The significance of the differences for $V_{1/2}$ values in Fig. 4, if any, needs to be evaluated using statistical methods rather than based on a visual inspection.

5. The conclusions drawn based on the analysis of amino acid substitutions for M155 and K464 residues in Fig. 4 and Fig. S2 are not clear. Qualitatively, it looks like the M155R mutation tends to shift the I/V plots to more hyperpolarized direction while M155E mutation causes a clearly visible shift to more depolarized direction (Fig. S2A), similar to the K464E mutation. In other words, it seems that the reduction of the positive charge between the interacting M155 and K464 residues seems to mimic the effect of cAMP or stabilize the open state. This suggests that not only the backbone at the position 155 but also the side-chain could be either directly or indirectly (allosterically) involved in M155/K464 interactions, while the authors conclude that primarily the backbone interactions at the 155 position are involved.

6. There seems to be a disconnect between MD simulations and predictions of interacting residues (Fig. 3A), interacting residues in Fig. 1B and functional results in Figs. S2B and S2C. A distance of less than 4 angstroms is typically required for a salt bridge formation. The authors state that E247 in the S2-S3 loop might be forming a salt bridge with K464, supposedly based on the MD simulations (Fig. 3A). Yet, Fig. 1B does not show a close enough proximity between these residues to promote a salt bridge formation. Therefore, the authors need to justify their assumption by indicating the distance measured between the residues in the HCN1 structure or the homology model, and including another figure, potentially in the supplement, showing the E247/K464 interaction from another angle that would indicate that these residues indeed are close enough to form a salt bridge. Without a proof of the structural possibility of the salt bridge formation, the rationale for considering E247A, E247R and E247/K464E mutations is not clear and the results of these mutations should be reinterpreted.

7. It is concerning that E247 and D244 were identified as interacting partners for K464 with MD simulations (Fig. 3A), yet, the experimental results do not support this prediction. In fact, the failure to rescue the wild-type phenotype with E247R_K464E mutation suggests that there is no salt bridge formation between these residues. This discrepancy between the MD prediction and functional results should be mentioned and possible explanations discussed.

8. The K464E mutant is still sensitive to cAMP, and cAMP causes shift of the I/V plots to more depolarized potentials (Fig. 2B). This is important as it suggests that cAMP is still binding to the mutant channel and is still increasing the channel activation. However, if the cAMP affinity is the same for the activated and non-activated K464E channels, as indicated by the patch-clamp fluorometry, it is not clear why is cAMP still shifting the I/V plots to more depolarized potentials. This potential discrepancy between the results of the patch-fluorometry and functional studies needs to be addressed in the discussion.

9. Although the functional data and MD simulations suggest that the structural changes due to K464E mutation are qualitatively similar to the ones induced by cAMP binding, there are substantial differences in the activation kinetics at more depolarized potentials between the K464E channels in the absence of cAMP and WT HCN channels in the presence of cAMP (Fig. 2C). This suggests that there are differences in the gating rearrangements between these two conditions. These differences should be acknowledged and the conclusions made based on the assumption that the cAMP-bound WT channels are similar to K464E channels need to be worded carefully.

Minor comments:

1. On pg. 15, second to last paragraph, it seems that K464 should be replaced by K464E.

2. Indicate SE for BC50 values in Fig. 5B.

Reviewer #3 (Remarks to the Author):

Review of Manuscript COMMSBIO-20-2796

Summary:

Based on previous structural and functional data for HCN ion channels, the authors raised the question of whether direct interactions between opposite subunits of HCN ion channels are essential for coupling conformational changes in the C-linker and cyclic nucleotide-binding (CNBD) domains (such as those promoted by cAMP binding) to ion channel gating. To address this question, the authors examined the mHCN2 channel using a combination of mutagenesis, electrophysiology, confocal patch-clamp fluorometry, and molecular dynamics (MD) simulations. In agreement with previous structures for hHCN1 and hHCN4, the authors determined that the C-linker forms interactions between opposite subunits of mHCN2 which are essential for stabilizing the closed state of the ion channel, and that a previously-identified rearrangement of the C-linker and CNBD domains associated with channel opening requires breakage or attenuation of these interactions. Furthermore, the authors identified C-linker residue K464 as a key amino acid residue for formation of the interactions, and confirmed that disrupting the interactions with K464 via mutation destabilizes the closed state of the ion channel. In addition, the authors found that CNBD perturbations occurring upon cAMP binding are also favored by breakage of the interactions with K464, which together with measured cAMP binding affinities, confirmed that cAMP binding is functionally coupled to the interactions with K464. Finally, the authors proposed a mechanism in which the previously-identified rearrangement of the C-linker and CNBD domains may promote channel opening by destabilizing the closed state of the channel via disruption of an interaction between the S5 and S6 helices.

Significance of Main Findings:

The authors identified previously under-appreciated direct interactions between opposite subunits within HCN ion channels that are involved in ion channel autoinhibition, and identified an amino acid residue key for formation of these interactions.

Strengths:

Overall, the article was well written, and followed a logical progression to address the question. The authors used a combination of multiple experimental (i.e. mutagenesis, electrophysiology, confocal patch-clamp fluorometry) and computational (i.e. structural modelling, MD simulations) analyses to arrive at a consensus for their results, comparing their results with previous structural data for further validation.

Weaknesses:

- 1.) The discussion within the Results subsection "Role of negatively charged residues in the S2-S3 loop for opposite subunit interactions" (pp. 13-14 of the article main text) was somewhat difficult to relate to the other results being presented, and possibly somewhat inconsistent with the corresponding text in the Discussion section. This Results subsection in its current state thus represents a possible weak point in the article.
- 2.) It should be pointed out that the smaller overall degree of structural shift (relative to the apo wild-type mHCN2 MD simulations) that was observed for the wild-type mHCN2 simulations with bound cAMP, compared to the apo K464E mHCN2 simulations (as exemplified in Figure 3B), may alternatively be due to inherent limitations of the unbiased MD simulation methodology. However, this does not seem to be acknowledged here.

3.) For the Results subsection "K464 acts via backbone but not via side-chain interaction with M155 of the opposite subunit", the authors could have better emphasized that the steady-state activation relationship shifts observed for the M155R and M155E mutants were shifts in directions opposite to the directions that would have been expected if K464 interacted with the side chain of M155, thereby ruling out interaction with the side chain of M155 as a likely contributor to the observed channel gating behavior.

Responses to the Reviewers

Manuscript: “Functional and structural characterization of interactions between opposite subunits in HCN channels”

Kondapuram and Frieg et al.

Reviewer #1 (Remarks to the Author):

Report on manuscript: Functional and structural characterization of interactions between opposite subunits in HCN pacemaker channels by Kondapuram and coworkers

The present study combines functional data, mutation analysis and computational studies for understanding the challenging question on how allosteric gating in HCN channels works. A pertinent question in the context of this gating mechanism is how the information of conformational changes are transduced from the ligand binding site into the channel part, which hosts the voltage sensitive gates. By scrutiny of the cryEM structure of the HCN1 channel the authors identify a conserved cationic amino acid in a critical position of the cytosolic domain, which might play a crucial role in this mechanical coupling. The experimental and computational studies indeed underpin a functional role of this cationic amino acid (K464). The data suggest that the Arg forms a hydrogen bond with the backbone of Met155 and that this coupling affects voltage and cAMP dependent gating.

As a side effect of the MD simulations the authors realized that K464 can also form salt bridges with anionic AA in the S2/S3 linker of the channel. In a mutant channel K464E this salt bridge formation is abolished with the effect that the CNBD rotates around the central axis of the channel. Since a similar rotation is seen between HCN structures with/without cAMP the authors speculate that this interaction could be an additional component in the mechanic transmission between ligand binding domain and channel gating.

The data are generally interesting and they show with no doubt functional of K464 in gating of HCN2. A positive aspect of the manuscript is the attempt to correlate functional/mutational data with potential conformational changes in the channel protein. But a critical evaluation of the data make me conclude that the data are heavily overinterpreted. In the context of published data (see below) I don't see how the entire mechanism of HCN2 gating can be interpreted in the context of a single amino acid. Also, while I find the computational data interesting they are too weak (see below) to make these strong points. For this reason I have to reject publication of the manuscript in the present form.

Major concerns:

1) The authors use MD simulations of a homology model based on the experimental HCN1 structure to examine atomic interactions of K464 within the channel protein. Since homology models bare the hazard of introducing structural artifacts and hence false positive interactions in a protein they should have performed a control simulation with the HCN1 structure. If as the authors imply the structural motive of interaction is universal in HCN channels they should have found the same interactions in the real structure. This would greatly support the impact of the computational data from the homology model.

4) The data in Fig. 3B are interpreted as evidence for a very small rotation or small rotation of the CNBD in presence of cAMP or in the K/E mutant respectively. I understand that this information is derived from the relative distance of the peaks of the distribution. If these narrow distributions with a small width I might understand this argument. But the distributions are very

broad, meaning that even in the mutant the orientation of the CNBD is not distinguishable from the wt.

Response: As to point 1, as suggested by the reviewer, we performed an analogous set of MD simulations for the hHCN1 channel. The atomic structure was solved by cryo-EM and served as the template for homology modeling of the mHCN2. Furthermore, and to address point 4, we adjusted the presentation of the results, in that we now show the CL-CNBD rotation relative to the starting structures as a boxplot with avg. rotation angle \pm SEM. As the mHCN2 and hHCN1 channels are homotetramers and 20 independent MD runs were performed, the mean values and SEMs were calculated for $n = 80$. Hence, the rotation angles describe the average rotation of one subunit relative to the starting structure (see further details below at point 3). For the mHCN2 channel, the results are shown in the revised Fig. 3B, and the results for hHCN1 are summarized in the new Fig. S4. Please note that we decided on the modified presentation as the reviewer wrote “*I understand that this information is derived from the relative distance of the peaks of the distribution*”, which was not the case but could be misjudged from our initial presentation.

To incorporate all changes stated above, we made several edits to the section “*K464 stabilizes the closed state by connecting two opposite subunits*” on pages 11-13 (lines 170-241).

Figure 3: Analyses of MD simulations of the wild type and K464E mHCN2 channel.

“... B: Average rotation angle relative to the channel pore in the starting structure. The direction of rotation is visualized by the scheme below the panels...”

“Figure S4: Analyses of MD simulations of the wild type and K422E hHCN1 channel.

A: A close-up view of the interaction site between two opposite subunits (colored cyan and yellow) in the hHCN1 channel (residues M94 to D608; based on PDB ID 5U6O (1) (used for MD simulations). M113, E201, D202, E205, and K422 (homologous residues to M155, E243, D244, E247, and K464 in the mHCN2 model) are depicted as sticks. The interaction between K422 and M113 is depicted as a gray dotted line. B: Average occurrence frequency of hydrogen bond interactions between two opposite subunits involving K422. K422 resides on subunit i , and the interaction partners M113, E201, D202, and E205 reside on subunit $i + 2$. As to M113, we only considered the backbone oxygen as H-bond acceptor; for E201, D202, and E205, we only considered the side-chain oxygen atoms, as we considered these interactions more favorable compared to backbone interactions. C: Average rotation angle relative to the channel pore in the starting structure. D: Average histogram (bin size 0.1°) of the rotation angle relative to the channel pore normalized to the apo wild type average (the vertical line at 0°). In C+D, the direction of rotation is visualized by the scheme below the panels. In B - D, the average values were calculated over all four subunits of the hHCN1 channel and 20 independent MD simulation runs ($n = 80$). The error bars denote the standard error of the mean (SEM). (p value by t -test; ** $p < 0.01$; n.s. not significantly different).”

2) The authors show that the occupation time of the hydrogen bond between K464 and its partners M155 or E247 is less than 25 %. This means, an interaction which is considered so crucial for structure/function relationship of HCN channels is much more absent than present. Notably in studies of other proteins the occupation time of critical salt bridges and/or hydrogen bonds is often in the order of 60 to 100%.

Response: With due respect, but we disagree with the reviewer’s point of criticism for the following reasons.

First, we consider the K464-mediated interactions as highly dynamic, such that rearrangements of interactions between K464 and M155, D244, and E247 are very likely. As stressed by the reviewer, only the K464-M155 interaction is present in the starting structure of these interactions. The S2-S3 linker is structurally not resolved in the cryo-EM structures of the hHCN1 (PDB ID: 5u6o, 6u6p), suggesting that this area is highly flexible, and we use a hierarchical approach (Jacobson et al., 2004 (DOI: 10.1002/prot.10613); Jacobson et al., 2002 (10.1016/S0022-2836(02)00470-9)) to incorporate this region during homology modeling of the mHCN2 channel. However, throughout MD simulations, K464-mediated interactions with M155, D244, and E247 break and (re-)form. In this regard, it is

astonishing that the cumulative occupation time of K464-mediated interactions is ~50% in mHCN2. The finding is further corroborated by our new simulation data on hHCN1, in which the cumulative occupation time of K422-mediated interactions is ~70%.

Second, our simulation data and functional *in vitro* data suggest that the rotation induced by K464E is not solely caused by the loss of a hydrogen bond to M155, D244, and E247 but, in particular, promoted by repulsive electrostatic forces with the S2-S3 linker, as we already wrote in the original manuscript. In the revised manuscript, the relevant section on pages 11-12 (lines 197-203) reads as

“Still, as the hydrogen bond between M155 and K464 is insensitive to cAMP binding (Figure 3A), one might assume that the rotation induced by K464E is not solely caused by the loss of a hydrogen bond to the backbone of M155. In line with this assumption, patch-clamp experiments revealed only small effects on the channel activation for the K464A mutant compared to the K464E mutant, also suggesting that the gating behavior in K464E is not solely caused by the loss of a hydrogen bond to the backbone of M155, which would also occur in the case of K464A.

The negatively charged residues E243, D244, and E247 on the S2/S3-linker are additional interaction partners of K464. In the apo wild type channel, only infrequent salt bridge interactions were found between K464 and E243 (in < 0.5 % of all conformations) or D244 (in 6.9 % ± 1.1 % of all conformations) (Figure 3A). A salt bridge interaction with E247 occurs in 20.1 % ± 2.4 % of all conformations (Figure 3A). Upon cAMP binding the interaction frequencies to D244 (in 2.7 % ± 0.7 % of all conformations; $p < 0.01$) and E247 (in 12.1 % ± 1.6 % of all conformations; $p < 0.01$) are significantly reduced (Figure 3A). Thus, it seems reasonable to assume that repulsive electrostatic forces, likely occurring in the K464E mutant, contribute to channel activation, which, in turn, mimics the reduced interaction frequencies observed upon cAMP binding. This assumption is supported by results from patch-clamp experiments, in which the activation voltage is shifted towards more positive values in the K464E mutant relative to the wild type channel, but to a lesser extent in K464A (Figure 2E, H).”

3) Furthermore, on the same issue, I do not understand whether this H⁺ bond frequency was calculated individually for each subunit or as a cumulative measure for all 4 subunits. In latter case the 25% frequency would have to be divided by 4. An interaction frequency of 5% for each subunit would not be more than a random event.

Response: All average values and error estimates were calculated for each subunit individually. To clarify this issue, we include this information in the figure caption. In the case of Fig. 3, for example, the caption (page 13) now reads: “ [...] In **A** and **B**, the average values were calculated individually for each of the four subunits and throughout 20 independent MD simulation replica ($n = 80$). The error bars denote the standard error of the mean (SEM). (p -value by t -test; * $p < 0.05$; ** $p < 0.01$; n.s. not significantly different).”

Additionally, in the Methods section entitled “Trajectory analysis” we now write: “Hydrogen bond interactions, RMSF values, distances, and rotation angles are reported as mean value ± SEM. As the mHCN2 channel is a homotetramer and as we performed 20 independent MD runs, the mean values and SEMs were calculated for $n = 80$ independent measurements, if not stated differently.”

5) The data in Fig. 3 imply that the K464 is able to interact with M155 and E247. If one of the interactions is eliminated by mutation of E247 one would expect that the frequency of interacting with M155 is favored. Is this observed in the MD simulations and reflected in the functional data?

Response: Elucidating the role of amino acids of the S2-S3 linker and how those residues are associated with cAMP-dependent gating in HCN2 channels was beyond the scope of this study, as it would require an independent set of MD simulations for each HCN2 variant. Considering the simulation time it took to generate the available simulation data on *apo* wild type, cAMP-bound, and *apo* K464E mHCN2 and the related hHCN1 states, it is impossible to address this question now. Our future research aims at such aspects.

6) The authors present their work as if they were the first to discover the importance of AA K464 and a functional interaction between the elbow and the CNBD in HCN channels. This is not the case. A simple search with the key word “K464 and HCN2” in Google shows that this AA was already mentioned by Pater et al. as a putative site for sumoylation (10.3389/fnmol.2016.00168). This is probably not important for the present analysis but it should be mentioned for completeness. More important however is the fact that Porro and coworkers (DOI: 10.7554/eLife.49672) already mention an interaction between M113-K422 in HCN1 (which is equivalent to M155 and K464 in HCN2). In their study individual and double mutations of K464 and E478 residues to alanines generated opposite effects on the voltage dependence of the channel. These data imply that the situation is not as straight forward as presented here.

Response: In the mentioned study of Parker and co-workers (2017) the authors investigate SUMOylation of HCN2 channels and its effect on surface expression and conductance. However, position K464 was not included in their study. The authors state that they have not tested if K464 is “involved in channel folding, assembly and/or trafficking, but it may be that SUMOylation plays a fundamental role in these processes for all HCN channels.” Because of this weak connection to our own manuscript, we decided not to add this paper to the references.

The paper of Porro and co-workers (2019) is mentioned in our original manuscript in the introduction and the discussion part. According to the statement of the reviewer, we now discuss this paper in more depth. However, herein, we are specifically investigating interactions between opposite subunits. Therefore, R154-E478 and other interactions between adjacent subunits were beyond the scope of our study.

To acknowledge the findings of Porro et al. (2019) and Wang et al. (2020) regarding adjacent subunits, we are referring now to interactions between the HCN domain and CL region of adjacent subunits in the INTRODUCTION section (page 4, lines 67-72) in the following context: *“Different techniques including electrophysiological approaches, fluorescence microscopy and cryo-electron microscopy combined with mutagenesis were used to show intensive interactions between neighboring subunits: S4-S5 linker-CL interactions (6, 11), CL-CL interactions (12-14), CNBD-CNBD interactions (6, 12), and very recently interactions between the newly discovered HCN domain with the CNBD, the voltage-sensing domain, and the CL region (6, 15, 16).”*

Regarding the type of interaction between M155 and K464 and regarding the supporting bond R154-E478 between adjacent subunits, we added the following to the DISCUSSION section (page 24-25, lines 464-481):

“An interaction between the backbone of M155 and the side chain of K464 was also suggested by Porro and co-workers (2019). They studied the mutant K464A, but not K464E, and observed a similar shift of $V_{1/2}$ under cAMP-free conditions as presented herein. However, they interpreted their data with the formation of a salt bridge and not a hydrogen bond as herein. They found that the cAMP effect is only lost after simultaneously breaking a second bond (E478-R154), which connects the C-linker with the HCN-domain of the adjacent subunit. In our simulations, the salt bridge between E478-R154 is insensitive to cAMP binding but significantly weakened in the K464E channel (Figure S8). However, the interaction frequencies are > 70% throughout all investigated channels and simulations,

suggesting that this interaction may be relevant for stabilizing adjacent subunits rather than being essential for cAMP-induced channel disinhibition. This assumption is supported by the functional data of Porro et al. on E478A HCN2, which is still highly sensitive to cAMP binding and almost identical to the wild type channel (Porro et al., 2019). How cAMP influences the R154A channel was not tested (Porro et al., 2019). Interestingly, the functional data on channel activation also suggests that the K464A-E478A HCN2 behaves identically to the apo wild type channel (Porro et al., 2019), while our K464E HCN2 mimics the cAMP-bound wild type channel (Figure 2). Our MD simulations suggest that the K464E channel adopts a conformation similar to the cAMP-bound wild type channel. One might speculate that K464A-E478A HCN2 adopts a conformation similar to the apo wild-type channel, which provides a plausible but not yet proven structural explanation for the functional data.”

Additionally, we added the following statement to the DISCUSSION section (page 27, lines 562-565): *“In summary, opposite subunits in HCN channels are functionally linked to each other. Our data suggest that the interactions between opposite subunits, embedded in a network of supporting interactions between adjacent subunits (Porro et al., 2019; Wang et al., 2020), substantially contribute to maintaining the autoinhibitory properties of HCN channels.”*

Not the same amino acids but a similar concept namely the interaction between C-linker and CNBD was also examined by Wang et al (doi: 10.1074/jbc.RA120.013281). In this study it was found that three residues on the C- linker/CNBD (E478, Q382 and H559) make direct interactions with residues R154 and S158 on the HCND. Disrupting these interactions affected both voltage- and cAMP-dependent gating of HCN2 channels. The critical AA are so close to the key residue of this study, K464, that these results cannot be ignored in the discussion of the present paper.

Response: We thank the reviewer for suggesting the paper of Wang and co-workers (2020). It is now included in the revised version of the manuscript both in the INTRODUCTION and in the DISCUSSION section. However, Wang and co-workers identified interactions between C-linker and HCN domain of adjacent subunits only. As already mentioned above, the aim of the present study was to identify interactions between opposite subunits. Interactions between adjacent subunits were beyond the scope of our study.

INTRODUCTION section page 4: see response to comment above

DISCUSSION section pages 24-25: see response to comment above

DISCUSSION section page 27: see response to comment above

Reviewer #2 (Remarks to the Author):

In this study, based on the analysis of the HCN1 channel structure and homology model of HCN2 channels, authors identified interactions between K464 on the C-linker of one subunit and M155 on the opposite subunit in HCN2 channels. K464E mutation shifted the current versus voltage plots to more depolarized potentials and decreased the cAMP sensitivity of the channel. MD simulations further supported direct interactions between M155 and K464 residues and suggested that K464E mutation introduces a rotation of the CL-CNBD in the same direction as induced by cAMP. Analysis of a cAMP fluorescent analog binding to the WT and K464E channels using patch clamp fluorometry indicated that the affinity of cAMP binding was the same for the activated (-130 mV) and non-activated (-30 mV) K464E mutant channels, and for the activated WT channels. Importantly, this binding affinity was almost two-fold higher than the cAMP affinity for WT non-activated channels. This suggests that the rotation of the CL-CNBD induced by K464E mutation increases cAMP binding affinity, supporting a hypothesis that voltage induced activation of HCN channels, also associated with similar rotation, increases cAMP binding affinity.

This is a very elegant study that for the first time focuses on examination of interactions between residues on the opposite subunits in HCN channels, while previous studies were focused on the interactions between residues on the adjacent subunits. The finding that the K464/M155 interaction favors the closed state of the channel, and destabilizing this interaction with K646E mutation favors an open state of the channel and mimics cAMP effect is interesting and important for understanding the gating mechanism of HCN channels. The data are high quality and findings are substantiated with complementary methods. However, there are some issues that need to be addressed.

Major comments:

1. Please include current traces for the mutant channels K464E and K464A in the absence and presence of cAMP.

Response: Current traces for K464E and K464A are shown now in a new Supplementary Figure S1. The subsequent Supplementary Figures have been renumbered accordingly.

Figure S1: Voltage-dependent activation of K464 mutants at zero and saturating [cAMP] (10 μ M). Representative current traces for K464E and K464A. The voltage protocol is illustrated above. Command voltages were applied in 10 mV-increments ranging from -70 mV to -150 mV.

2. Although K464E mutation decreased the differences in the current versus voltage (I/V) plots and activation kinetics in the absence and presence of cAMP, and brought these values closer to the ones for WT HCN channels in the presence of cAMP, Figs. 2B and 2C show noticeable differences. Therefore, phrases “largely superimposed” and “largely similar” should be avoided, unless there is a rigorous statistical analysis done to indicate that the data in the absence and presence of cAMP are not statistically different.

Response: We agree with the reviewers concern about the phrases “largely superimposed” and “largely similar” and changed the text and Figure 4 accordingly. Because we reduced the information in the new Figure 2G to -120 mV only, we added a new Supplementary Figure S2 showing deactivation kinetics as in the original manuscript. The subsequent Supplementary Figures have been renumbered accordingly.

RESULTS section, Page 8 (lines 128-134), describing the steady-state parameter $V_{1/2}$: “As expected from the literature, in HCN2 wild type channels the application of cAMP led to a pronounced shift of the steady-state activation relationship to more depolarized voltages ($\Delta V_{1/2} = 17.9 \pm 1.1$ mV (Figure 2B,H). Interestingly, the relationship for K464E in the absence of cAMP resembled the cAMP-saturated mHCN2 wild type channel by causing a $V_{1/2}$ value of -96.1 ± 1.0 mV compared to a $V_{1/2}$ value of -100.6 ± 1.3 mV. Hence, adding saturating cAMP concentrations shifted the relationship by a minor but significant extent of 5.24 ± 1.1 mV to more depolarized voltages (Figure 2B,H).”

RESULTS section, Page 8 (lines 135-147), describing the activation kinetics: “To evaluate the activation kinetics, we fitted an exponential function (eq. (2)) to the time courses of activating currents, yielding the time constant of activation, τ_{act} . The results are summarized in Figure 2C. Because each construct was responsive to a different range of command voltages, resulting in different activation states for each voltage, we decided to plot the activation time constants τ_{act} not only versus the command voltage (Figure 2C), but additionally versus the normalized voltage $V_{command}/V_{1/2}$ (Figure 2). For control conditions in the absence of cAMP, and for high activation states in the presence of cAMP, there was no difference in activation kinetics for K464E and wild type mHCN2. In contrast, K464E activation kinetics was accelerated in comparison to HCN2 activation kinetics for lower activation states when cAMP was bound. However, due to the limited time window of the activating pulse and the lower current amplitudes in this voltage range, those differences should be interpreted carefully. Together, the activation kinetics suggests that rate-limiting steps for channel activation were not substantially affected by the mutation K464E.”

RESULTS section, Page 8 (lines 151-153), describing the deactivation kinetics: “Because τ_{deact} was independent from the activating voltage pulse, only values obtained from -120 mV pulses are shown in Figure 2D (for τ_{deact} values at the whole voltage range see Supplement Figure S2).

“Figure 1: Voltage-dependent activation of K464 mutants at zero and saturating [cAMP] (10 μ M).

A: Exemplary current traces for mHCN2 activation. Left panel: protocol and representative traces for measuring steady-state activation and activation kinetics at zero [cAMP], right panel: as left panel but at saturating [cAMP] (10 μ M). **B, E:** Steady-state activation for K464E and K464A at zero and saturating [cAMP] in comparison to mHCN2, respectively. Solid lines indicate the result of a Boltzmann fit (equation 1) ($n = 5$ to 9). **C, F:** Activation kinetics at zero and saturating [cAMP] for K464E, K464A, and mHCN2 ($n = 5$ to 9). **D:** Activation kinetics for mHCN2 and K464E plotted versus normalized command voltage ($V_{\text{command}}/V_{1/2}$), **G:** Deactivation kinetics at zero and saturating [cAMP] for K464E, K464A, and mHCN2 after -120 mV command voltage ($n = 5$ to 7)....”

“Figure S2: Deactivation kinetics for K464 mutants.

Deactivation time constants at zero and saturating [cAMP] for K464E (A) and K464A (B) in comparison to mHCN2 ($n = 5$ to 7).”

3. Although the authors focus on the interactions between M113 on the HCN domain of one subunit and K464 on the C-linker of the opposing subunits, Wang et al. (JBC, 2020) investigated interactions between R112 on the HCN domain of one subunit and E478/Q478 on the C-linker of the adjacent subunit, and Porro et al (2019) considered interactions between R112 and E478 from the adjacent subunit. R112 and M113 are adjacent residues and the global conformational changes happening during channel activation are bound to affect interactions between both R112 and M113. The proximity of the interactions between the opposite and adjacent subunits indicates the strategic location of these residues important for the channel gating. These previous findings should be discussed in the manuscript.

Response: We follow the reviewers suggestion and discuss the papers of Wang et al. (2010) and Porro et al. (2020) in more depth. We added the following paragraphs to the revised manuscript:

The paper of Porro and co-workers (2019) is mentioned in our original manuscript in the introduction and the discussion part. According to the statement of the reviewer, we now discuss this paper in more depth. However, herein, we are specifically investigating interactions between opposite subunits. Therefore, R154-E478 and other interactions between adjacent subunits were beyond the scope of our study.

To acknowledge the findings of Porro et al. (2019) and Wang et al. (2020) regarding adjacent subunits, we are referring now to interactions between the HCN domain and CL region of adjacent subunits in the INTRODUCTION section (page 4, lines 67-72) in the following context: *“Different techniques including electrophysiological approaches, fluorescence microscopy and cryo-electron microscopy combined with mutagenesis were used to show intensive interactions between neighboring subunits: S4-S5 linker-CL interactions (6, 11), CL-CL interactions (12-14), CNBD-CNBD interactions (6, 12), and very recently interactions between the newly discovered HCN domain with the CNBD, the voltage-sensing domain, and the CL region (6, 15, 16).”*

Regarding the type of interaction between M155 and K464 and regarding the supporting bond R154-E478 between adjacent subunits, we added the following to the DISCUSSION section (page 24, lines 464-481):

“An interaction between the backbone of M155 and the side chain of K464 was also suggested by Porro and co-workers (2019). They studied the mutant K464A, but not K464E, and observed a similar shift of $V_{1/2}$ under cAMP-free conditions as presented herein. However, they interpreted their data with the formation of a salt bridge and not a hydrogen

bond as herein. They found that the cAMP effect is only lost after simultaneously breaking a second bond (E478-R154), which connects the C-linker with the HCN-domain of the adjacent subunit. In our simulations, the salt bridge between E478-R154 is insensitive to cAMP binding but significantly weakened in the K464E channel (Figure S8). However, the interaction frequencies are > 70% throughout all investigated channels and simulations, suggesting that this interaction may be relevant for stabilizing adjacent subunits rather than being essential for cAMP-induced channel disinhibition. This assumption is supported by the functional data of Porro et al. on E478A HCN2, which is still highly sensitive to cAMP binding and almost identical to the wild type channel (Porro et al., 2019). How cAMP influences the R154A channel was not tested (Porro et al., 2019). Interestingly, the functional data on channel activation also suggests that the K464A-E478A HCN2 behaves identically to the apo wild type channel (Porro et al., 2019), while our K464E HCN2 mimics the cAMP-bound wild type channel (Figure 2). Our MD simulations suggest that the K464E channel adopts a conformation similar to the cAMP-bound wild type channel. One might speculate that K464A-E478A HCN2 adopts a conformation similar to the apo wild-type channel, which provides a plausible but not yet proven structural explanation for the functional data.”

Additionally, we added the following statement to the DISCUSSION section (page 27, lines 562-565): “In summary, opposite subunits in HCN channels are functionally linked to each other. Our data suggest that the interactions between opposite subunits, embedded in a network of supporting interactions between adjacent subunits (Porro et al., 2019; Wang et al., 2020), substantially contribute to maintaining the autoinhibitory properties of HCN channels.”

4. The significance of the differences for $V_{1/2}$ values in Fig. 4, if any, needs to be evaluated using statistical methods rather than based on a visual inspection.

Response: We agree with the reviewer that the statistical evaluation of the data presented in Figure 4 is missing in the original manuscript. We performed such an evaluation by applying the students *t*-test and present the results as symbols in Figure 4. In parallel, we rephrased the respective text paragraphs and mention now the results of the significance tests also verbally in the RESULTS section.

For position M155, it reads now as follows (page 14, lines 252-259): “M155A showed no difference in the steady-state activation relationship compared to mHCN2, neither in the absence of cAMP nor in the presence of saturating [cAMP] (Figure 4A, Figure S5), resulting in a similar $\Delta V_{1/2}$. In M155R, the relationship in the presence of cAMP was slightly shifted to more hyperpolarized voltages while it was similar to mHCN2 in the absence of cAMP. However, the cAMP-induced $\Delta V_{1/2}$ was in the range of wild type mHCN2. In M155E, both curves were shifted to more depolarized voltages, indicating an effect on voltage-dependent gating and a stabilization of the open state. However, there was no significant effect on $\Delta V_{1/2}$; thus, the cAMP-dependent gating was not affected.

For position E247, it reads now as follows (page 15-16, lines 272-281): “As described above, E247 in the S2-S3 linker was identified as a potential interaction partner for K464 (besides M155) to form a salt bridge. Thus, we constructed the mutants E247A and E247R to study the role of this residue for channel gating. The data are summarized in Figure 4B. Surprisingly, cAMP-induced gating is not affected. The cAMP-induced shift of $V_{1/2}$ is similar to the shift shown for wild type channels. However, there is a significant effect on voltage-induced gating: The steady-state activation relationships were shifted to more negative voltages for both mutants, indicating a stabilization of the closed state. From this, it can be concluded that changing the charge at position 247 in the S2-S3 linker, that way breaking

a potential bond between E247 and K464, has no negative effect on the bond between K464 and M155.”

For position 244, it reads now as follows (page 16, lines 286-293): “Additionally, our MD data showed a low probability of forming salt bridge interactions between K464 and D244 (Figure 3A). We tested the role of D244 for channel gating by constructing D244A and D244K. In both cases, $V_{1/2}$ values were shifted to more negative values in the absence of cAMP, while there were no changes for cAMP-saturated constructs (Figure 4C). Consequently, $\Delta V_{1/2}$ was significantly increased rather than decreased as shown for K464E. From this, it can be concluded that changing the charge at position 244 in the S2-S3 linker, that way breaking a potential bond between E244 and K464, has no negative effect on the bond between K464 and M155.”

“...Asterisks indicate significant differences for comparison with mHCN2 at the respective cAMP condition (Student’s *t*-test ($p < 0.05$)) or for the $\Delta V_{1/2}$ values.”

5. The conclusions drawn based on the analysis of amino acid substitutions for M155 and K464 residues in Fig. 4 and Fig. S2 are not clear. Qualitatively, it looks like the M155R mutation tends to shift the *I/V* plots to more hyperpolarized direction while M155E mutation causes a clearly visible shift to more depolarized direction (Fig. S2A), similar to the K464E mutation. In other words, it seems that the reduction of the positive charge between the interacting M155 and K464 residues seems to mimic the effect of cAMP or stabilize the open state. This suggests that not only the backbone at the position 155 but also the side-chain could be either directly or indirectly (allosterically) involved in M155/K464 interactions, while the authors conclude that primarily the backbone interactions at the 155 position are involved.

Response: We agree with the reviewer that the voltage-dependent gating of M155R and M155E differs. For details see responses to comment 4 and revised Figure 4, which indicates now significant differences. While the *I-V* relationship for M155R at saturating cAMP was shifted towards hyperpolarized voltages, both *I-V* relationships for M155E were

significantly shifted towards depolarized voltages. However, for both mutants, $\Delta V_{1/2}$ was not different from wild type mHCN2.

If there was a possibility of the side chains of position 155 and 464 to react, a negative glutamate side chain at position 155 and a positive lysin side chain at position 464 would most likely form a salt bridge. Thus, a stabilizing bond between HCN domains and C-linkers of opposite subunits would be created. Considering that the C-linker rotates away from the HCN domain to open the channel, such a bond would rather stabilize the closed state than the open state, as suggested by the steady-state activation data from M155E. From this, we conclude that K464 is not acting via a side chain interaction with the residue at position 155.

For more clarity, we rephrased the last paragraph on page 14 (lines 262-269): *“Neither shortening the side-chain at position 155 (M155A) nor adding a positive charge (M155R) to test for repulsive forces between position M155 and K464 affected the gating behavior in a similar way as did K464E, which supports our idea that the side chain at position 155 is not interacting with the side chain of K464. Moreover, in M155E, analysis of steady-state activation revealed an open state-stabilization. If the introduced glutamate side chain interacted with the K464 side chain, most likely by forming a stabilizing salt bridge between HCN domain and opposite C-linker, we would rather expect a stabilization of the closed state.”*

6. There seems to be a disconnect between MD simulations and predictions of interacting residues (Fig. 3A), interacting residues in Fig. 1B and functional results in Figs. S2B and S2C. A distance of less than 4 angstroms is typically required for a salt bridge formation. The authors state that E247 in the S2-S3 loop might be forming a salt bridge with K464, supposedly based on the MD simulations (Fig. 3A). Yet, Fig. 1B does not show a close enough proximity between these residues to promote a salt bridge formation. Therefore, the authors need to justify their assumption by indicating the distance measured between the residues in the HCN1 structure or the homology model, and including another figure, potentially in the supplement, showing the E247/K464 interaction from another angle that would indicate that these residues indeed are close enough to form a salt bridge. Without a proof of the structural possibility of the salt bridge formation, the rationale for considering E247A, E247R and E247/K464E mutations is not clear and the results of these mutations should be reinterpreted.

Response: We consider the K464-mediated interactions as highly dynamic, such that rearrangements of interaction between K464 and M155, D244, and E247 are very likely. As stressed by the reviewer, only the K464-M155 interaction is present in the starting structure from these interactions. The S2/S3-linker is structurally not resolved in cryo-EM structures of the hHCN1 (PDB: 5u6o, 6u6p), suggesting that this area is highly flexible, and we use a hierarchical approach (DOI: 10.1002/prot.10613; 10.1016/S0022-2836(02)00470-9) to incorporate this region during homology modeling of the mHCN2 channel. However, throughout MD simulations, K464-mediated interactions with M155, D244, and E247 break and (re-)form. For evaluation whether a hydrogen bond exists, we used common geometric descriptions of hydrogen bonds, which we state in the Methods section on page 32 (lines 719-723) and reads as:

“To investigate the interactions between two opposite subunits, we measured hydrogen bond interactions between K464 from one subunit to M155, E243, D244, and E247 from the other subunit. Hydrogen bond interactions were determined using a distance of 3.5 Å between the two donor and acceptor atoms and an angle (donor atom, H, acceptor atom) of 120° as cutoff criteria (20).”

Consequently, if at least one of the above-mentioned geometric criteria is not met during the simulations, this conformation was not counted as an interaction. Of course, the system is very dynamic, allowing the breaking and (re-)formation of interactions.

7. It is concerning that E247 and D244 were identified as interacting partners for K464 with MD simulations (Fig. 3A), yet, the experimental results do not support this prediction. In fact, the failure to rescue the wild type phenotype with the E247R_K464E mutation suggests that there is no salt bridge formation between these residues. This discrepancy between the MD prediction and functional results should be mentioned and possible explanations discussed.

Response: We agree with the reviewer's concern that our functional data did not support the idea of a single bond between K464 and either E247 or D244. We therefore state already in our original manuscript, that the function of K464 in mHCN2 wild type does not require the formation of a single interaction with either E247 or D244, but that instead D244 or E247 take over the role of an interaction partner of K464 if the respective other residue is substituted (page 25).

8. The K464E mutant is still sensitive to cAMP, and cAMP causes a shift of the I/V plots to more depolarized potentials (Fig. 2B). This is important as it suggests that cAMP is still binding to the mutant channel and is still increasing the channel activation. However, if the cAMP affinity is the same for the activated and non-activated K464E channels, as indicated by the patch-clamp fluorometry, it is not clear why is cAMP still shifting the I/V plots to more depolarized potentials. This potential discrepancy between the results of the patch-fluorometry and functional studies needs to be addressed in the discussion.

Response: We disagree with the reviewer that there is a discrepancy between the results of the patch-clamp fluorometry and electrophysiological data. As the reviewer mentioned correctly in his/her comment, from both sets of data it is obvious that cAMP binds to K464E channels. From electrophysiological data, we learned that the ability of cAMP to stabilize the open state is reduced in K464E because the open state of empty K464E channels is already stabilized compared to wild type mHCN2.

Together with our own MD data and data from previous publications, we concluded that this open-state stabilization is caused by a leftward rotated position of the CL-disk in non-activated channels compared to the position in non-activated wild type channels. The high affinity in non-activated K464E, seen in our PCF recordings, is most likely initiated by this rotated CL-linker. However, this fact does not exclude that cAMP still affects channel gating via conformational changes besides CL rotation.

9. Although the functional data and MD simulations suggest that the structural changes due to K464E mutation are qualitatively similar to the ones induced by cAMP binding, there are substantial differences in the activation kinetics at more depolarized potentials between the K464E channels in the absence of cAMP and WT HCN channels in the presence of cAMP (Fig. 2C). This suggests that there are differences in the gating rearrangements between these two conditions. These differences should be acknowledged and the conclusions made based on the assumption that the cAMP-bound WT channels are similar to K464E channels need to be worded carefully.

Response: We follow the reviewer's suggestion and now discuss the activation kinetics in more detail. We also revised Figure 4 for the sake of more clarity. For the text changes, see our response to comment 2.

Minor comments:

1. On pg. 15, second to last paragraph, it seems that K464 should be replaced by K464E.

Response: We corrected two cases accordingly.

2. Indicate SE for BC50 values in Fig. 5B.

Response: We assume the reviewer is referring to Figure 5C. We added the following sentence to the figure legend: "*Error bars indicate SEM.*"

Reviewer #3 (Remarks to the Author):

Review of Manuscript COMMSBIO-20-2796

Summary:

Based on previous structural and functional data for HCN ion channels, the authors raised the question of whether direct interactions between opposite subunits of HCN ion channels are essential for coupling conformational changes in the C-linker and cyclic nucleotide-binding (CNBD) domains (such as those promoted by cAMP binding) to ion channel gating. To address this question, the authors examined the mHCN2 channel using a combination of mutagenesis, electrophysiology, confocal patch-clamp fluorometry, and molecular dynamics (MD) simulations. In agreement with previous structures for hHCN1 and hHCN4, the authors determined that the C-linker forms interactions between opposite subunits of mHCN2 which are essential for stabilizing the closed state of the ion channel, and that a previously-identified rearrangement of the C-linker and CNBD domains associated with channel opening requires breakage or attenuation of these interactions. Furthermore, the authors identified C-linker residue K464 as a key amino acid residue for formation of the interactions, and confirmed that disrupting the interactions with K464 via mutation destabilizes the closed state of the ion channel. In addition, the authors found that CNBD perturbations occurring upon cAMP binding are also favored by breakage of the interactions with K464, which together with measured cAMP binding affinities, confirmed that cAMP binding is functionally coupled to the interactions with K464. Finally, the authors proposed a mechanism in which the previously-identified rearrangement of the C-linker and CNBD domains may promote channel opening by destabilizing the closed state of the channel via disruption of an interaction between the S5 and S6 helices.

Significance of Main Findings:

The authors identified previously under-appreciated direct interactions between opposite subunits within HCN ion channels that are involved in ion channel autoinhibition, and identified an amino acid residue key for formation of these interactions.

Strengths:

Overall, the article was well written, and followed a logical progression to address the question. The authors used a combination of multiple experimental (i.e. mutagenesis, electrophysiology, confocal patch-clamp fluorometry) and computational (i.e. structural modelling, MD simulations) analyses to arrive at a consensus for their results, comparing their results with previous structural data for further validation.

Weaknesses:

1.) The discussion within the Results subsection "Role of negatively charged residues in the S2-S3 loop for opposite subunit interactions" (pp. 13-14 of the article main text) was somewhat difficult to relate to the other results being presented, and possibly somewhat inconsistent with the corresponding text in the Discussion section. This Results subsection in its current state thus represents a possible weak point in the article.

Response: We thank the reviewer for this comment and rephrased the paragraph "Role of negatively charged residues in the S2-S3 linker for opposite subunit interactions" in the RESULTS section (page 15,16, lines 272-293). It reads now:

"As described above, E247 in the S2-S3 linker was identified as a potential interaction partner for K464 (besides M155) to form a salt bridge. Thus, we constructed the mutants E247A and E247R to study the role of this residue for channel gating. The data are summarized in Figure 4B. Surprisingly, cAMP-induced gating is not affected. The cAMP-induced shift of $V_{1/2}$ is similar to the shift shown for wild type channels. However, there is a significant effect on voltage-induced gating: The steady-state activation relationships were

shifted to more negative voltages for both mutants, indicating a stabilization of the closed state. From this, it can be concluded that changing the charge at position 247 in the S2-S3 linker, that way breaking a potential bond between E247 and K464, has no negative effect on the bond between K464 and M155. This result is further supported by the gating behavior of the mutant E247R_K464E. If a salt bridge between K464 and E247 mediated the function of K464, such a bond should be rescued in E247R_K464E, leading to a wild type-like phenotype. However, in E247R_K464E, the cAMP dependence was still strongly reduced like in K464E ($V_{1/2} = 4.2 \pm 1.2$ mV) (Figure 4B).

Additionally, our MD data showed a low probability of forming salt bridge interactions between K464 and D244 (Figure 3A). We tested the role of D244 for channel gating by constructing D244A and D244K. In both cases, $V_{1/2}$ values were shifted to more negative values in the absence of cAMP, while there were no changes for cAMP-saturated constructs (Figure 4C). Consequently, $\Delta V_{1/2}$ was significantly increased rather than decreased as shown for K464E. From this, it can be concluded that changing the charge at position 244 in the S2-S3 linker, that way breaking a potential bond between E244 and K464, has no negative effect on the bond between K464 and M155.”

We also adapted the respective paragraph in the DISCUSSION section (page 25, lines 489-504):

“We next studied two negatively charged residues in the opposite S2-S3 linker, D244 and E247. Neither neutralization nor charge reversal at those positions resulted in phenotypes similar to K464A or K464E. We thus concluded that the function of K464 in mHCN2 wild type does not rely on the formation of a bond with either one of those residues. To further support this conclusion for the residue E247, which showed the highest likelihood of developing a salt bridge with K464, we confirmed with the mutant E247R_K464E that rescuing a possible salt bridge does not result in a wild type-like phenotype. This construct still behaved like the K464E mutant, possibly because the proposed repulsive forces introduced by the glutamate at position K464 still find another counterpart in other negatively charged residues near position 247, for instance, D244. This finding may further suggest that D244 or E247 take over the role of an interaction partner of K464 if the respective other residue is substituted.

Notably, all four constructs showed a shift of $V_{1/2}$ to more negative values in the absence of cAMP, indicating a stabilization of the closed state. Because $V_{1/2}$ at saturating cAMP is less affected, for all constructs, the cAMP-induced $\Delta V_{1/2}$ is more pronounced than for mHCN2 wild type. Thus, the negatively charged residues in the S2-S3 linker seem to be involved in voltage-dependent rather than in cAMP-dependent gating.”

2.) It should be pointed out that the smaller overall degree of structural shift (relative to the apo wild type mHCN2 MD simulations) that was observed for the wild type mHCN2 simulations with bound cAMP, compared to the apo K464E mHCN2 simulations (as exemplified in Figure 3B), may alternatively be due to inherent limitations of the unbiased MD simulation methodology. However, this does not seem to be acknowledged here.

Response: As suggested, we now include such a paragraph in the Discussion section (pages 24 and 25, lines 455-481), such that it reads: “One might argue that the small differences between the apo K464E mHCN2 simulations may be due to inherent limitations of the unbiased MD simulation. However, we followed the extensively validated “ensemble average approach”, a procedure to interpret equilibrium dynamics from multiple, independent MD trajectories (Loccisano et al., 2004 (DOI: 10.1016/j.jmgm.2003.12.004); Likić et al., 2005 (DOI: 10.1110/ps.051681605); Caves et al., 2008 (DOI: 10.1002/pro.5560070314), to minimize the potential impact of insufficient sampling.

Moreover, we started the MD simulations from the same channel conformation, allowing to interpret results between the mHCN2 simulations on a relative basis.”

3.) For the Results subsection “K464 acts via backbone but not via side-chain interaction with M155 of the opposite subunit”, the authors could have better emphasized that the steady-state activation relationship shifts observed for the M155R and M155E mutants were shifts in directions opposite to the directions that would have been expected if K464 interacted with the side chain of M155, thereby ruling out interaction with the side chain of M155 as a likely contributor to the observed channel gating behavior.

Response: We thank the reviewer for this comment and rephrased the respective paragraph in the RESULTS section (page 14, lines 262-269).

“Neither shortening the side-chain at position 155 (M155A) nor adding a positive charge (M155R) to test for repulsive forces between position M155 and K464 affected the gating behavior in a similar way as did K464E, which supports our idea that the side chain at position 155 is not interacting directly with the side chain of K464. Moreover, in M155E, analysis of steady-state activation revealed an open state-stabilization. If the introduced glutamate side chain interacted with the K464 side chain, most likely by forming a stabilizing salt bridge between HCN domain and opposite C-linker, we rather would expect a stabilization of the closed state.”

Reviewers' comments:

Reviewer #1 (Remarks to the Author):

I appreciate the effort of the Authors to improve the manuscript. But I regret to say that I still find the data interesting but overinterpreted with respect to a general mechanism in HCN channels. The proposed mechanism relies entirely on the phenotype of one mutation and the general mechanism is not supported by the data.

The Authors present their hypothesis that K464 makes a crucial interaction with M155 throughout the paper, but the data do not seem to support this picture:

Hypothesis of authors: The interaction of K464 with M155 is crucial for cAMP mediated gating, in the sense that it is lost upon cAMP binding.

Test of the hypothesis: Arg464 was neutralized into Ala. This eliminates the crucial H-bond intramolecular interaction.

Results: The mutation has no effect on channel function and cAMP dependency.

My resume: the interaction between K464 and M155 cannot be as crucial as expected.

The authors then find that a substitution of Arg464 with an anionic amino acid has an effect on the voltage dependency in that it shifts the activation curve to positive voltages.

Hypothesis of authors: The electro-repulsion, which is induced by this mutation, mimics the cAMP bound conformation of the protein.

Test of the hypothesis: cAMP is added to the mutant channel. If the mutant would represent the cAMP bound conformation of the protein one would expect no further effect of cAMP. Result: The data show that cAMP is still shifting the activation curve of the mutant, even more to the right.

My resume: the two effects are additive, i.e. the two mechanisms (that of the mutation and that of cAMP) are independent.

The authors analyze the effects of the K464E mutation by MD simulation of a homology model of HCN2.

Hypothesis of authors: They assume that the mutant conformation resembles that of the ligand bound protein. The expectation is that the apo-mutant protein and the holo-wt protein should behave the same.

Test of hypothesis: Comparison of the two proteins by MD shows that the ligand has no significant effect on the propensity of hydrogen bond formation between K464 and M155. Also, the ligand causes only a very small, if any, rotation of the CNBD. The mutation on the other hand generates a strong reduction in hydrogen bond formation and a visible albeit still small rotation of the CNBD. At this point it should be mentioned that the 2.8° angle rotation is more similar to that of the structures of HCN1 (1° angle) that does not respond to cAMP, but very different from that of the hHCN4 (about 8°) that responds to cAMP.

My interpretation: the mutant is not reproducing the effect of the ligand. If anything, the mutant is evoking structural changes, which go beyond that of the ligand. But as we know from the functional data: the ligand is even adding a shift in the activation curve to the mutant. Hence the mutant cannot have a stronger effect than the ligand.

Hypothesis: The authors conclude from the MD simulations that K464 interacts with M155A.

Test of hypothesis: Methionine has been mutated into Ala, Arg, and Glu.

Results: These mutations are not abolishing the cAMP dependency of the channel. The authors interpret this as evidence for an interaction of K464 with the amino acid backbone.

My interpretation: as a devil advocate, I could also conclude from these data that the results of the MD simulation are not validated and that K464 does not at all interact with M155. The absence of a functional effect is not a strong argument for the proposed interaction. This critical statement is further supported by the fact that the insertion of a cationic or anionic amino acid in this position has no impact on channel function. If we consider the long side chains of the cationic and anionic

amino acids it seems even surprising that they do not interact with the cationic K464.

The authors find in the MD simulations that K464 does not only form H+ bonds with M155 but also, with a lower propensity, salt bridges with adjacent anionic amino acids.

Hypothesis: If the data from the MD simulation on H bond formation are correct also the salt bridge formations should be structurally relevant. This is in particular the case since salt bridges have in general a much higher energy than H-bonds.

Test of the hypothesis. Mutations, which eliminate the respective salt bridges were tested for cAMP sensitivity.

Result: the mutations have no appreciable effect on function.

My interpretation: it is at least possible that the predictions from the MD simulations are not correct. We have to consider here that the model is indeed only a homology model even though it is based on an experimentally resolved structure of the closely related HCN1 channel. But at this point it is important to remember that some of the structures, which are critical in this manuscript, such as the S2-S3 loop, are not even resolved properly in the HCN1 structure and they had to be modelled anew. They were modelled on the basis of general structural rules for proteins. This bears the hazard of artifacts in that the real structure may be different from the modeled one. To exemplify this, I noticed that the Authors used in one of the gap filling models the cAMP bound structure of the CNBD from HCN1 to model the missing structures of the C and D helices in the apo. This seems to me a wrong maneuver since these domains undergo significant folding only after cAMP binding. This is way they are not present in the original apo structure of HCN1. Hence modeling this part of the apo HCN2 channel, guided by structures from the cAMP bound HCN1, could introduce serious artifacts.

Reviewer #2 (Remarks to the Author):

The authors addressed the reviewer's concerns adequately and incorporated additional explanations and discussion. I have no further comments.

Reviewer #3 (Remarks to the Author):

Review of Revised Manuscript COMMSBIO-20-2796A: Reply to Author Revisions/Rebuttal of COMMSBIO-20-2796 Comments

Author Responses to COMMSBIO-20-2796 Comments:

Comment #1: The discussion within the Results subsection "Role of negatively charged residues in the S2-S3 loop for opposite subunit interactions" (pp. 13-14 of the article main text) was somewhat difficult to relate to the other results being presented, and possibly somewhat inconsistent with the corresponding text in the Discussion section. This Results subsection in its current state thus represents a possible weak point in the article.

Author Response: We thank the reviewer for this comment and rephrased the paragraph "Role of negatively charged residues in the S2-S3 linker for opposite subunit interactions" in the RESULTS section (page 15,16, lines 272-293). It reads now:
"As described above, E247 in the S2-S3 linker was identified as a potential interaction partner for K464 (besides M155) to form a salt bridge. Thus, we constructed the mutants E247A and E247R to study the role of this residue for channel gating. The data are summarized in Figure 4B. Surprisingly, cAMP-induced gating is not affected. The cAMP-induced shift of $V_{1/2}$ is similar to the shift shown for wild type channels. However, there is a significant effect on voltage-induced gating: The steady-state activation relationships were shifted to more negative voltages for both mutants, indicating a stabilization of the closed state. From this, it can be concluded that changing the charge at position 247 in the S2-S3 linker, that way breaking a potential bond between E247 and K464, has no negative effect on the bond between K464 and M155. This result is further

supported by the gating behavior of the mutant E247R_K464E. If a salt bridge between K464 and E247 mediated the function of K464, such a bond should be rescued in E247R_K464E, leading to a wild type-like phenotype. However, in E247R_K464E, the cAMP dependence was still strongly reduced like in K464E ($V_{1/2} = 4.2 \pm 1.2$ mV) (Figure 4B).

Additionally, our MD data showed a low probability of forming salt bridge interactions between K464 and D244 (Figure 3A). We tested the role of D244 for channel gating by constructing D244A and D244K. In both cases, $V_{1/2}$ values were shifted to more negative values in the absence of cAMP, while there were no changes for cAMP-saturated constructs (Figure 4C). Consequently, $\Delta V_{1/2}$ was significantly increased rather than decreased as shown for K464E. From this, it can be concluded that changing the charge at position 244 in the S2-S3 linker, that way breaking a potential bond between E244 and K464, has no negative effect on the bond between K464 and M155."

We also adapted the respective paragraph in the DISCUSSION section (page 25, lines 489-504): "We next studied two negatively charged residues in the opposite S2-S3 linker, D244 and E247. Neither neutralization nor charge reversal at those positions resulted in phenotypes similar to K464A or K464E. We thus concluded that the function of K464 in mHCN2 wild type does not rely on the formation of a bond with either one of those residues. To further support this conclusion for the residue E247, which showed the highest likelihood of developing a salt bridge with K464, we confirmed with the mutant E247R_K464E that rescuing a possible salt bridge does not result in a wild type-like phenotype. This construct still behaved like the K464E mutant, possibly because the proposed repulsive forces introduced by the glutamate at position K464 still find another counterpart in other negatively charged residues near position 247, for instance, D244. This finding may further suggest that D244 or E247 take over the role of an interaction partner of K464 if the respective other residue is substituted.

Notably, all four constructs showed a shift of $V_{1/2}$ to more negative values in the absence of cAMP, indicating a stabilization of the closed state. Because $V_{1/2}$ at saturating cAMP is less affected, for all constructs, the cAMP-induced $\Delta V_{1/2}$ is more pronounced than for mHCN2 wild type. Thus, the negatively charged residues in the S2-S3 linker seem to be involved in voltage-dependent rather than in cAMP-dependent gating."

==> Reply to Author Response:

In the revised "Role of negatively charged residues in the S2-S3 linker for opposite subunit interactions" RESULTS subsection (page 15,16, lines 272-293), the phrase "has no negative effect on the bond between K464 and M155" seems somewhat out of place here, although I understand that this text was also added to try to address comments from the other reviewers. I suggest changing the phrase "has no negative effect on the bond between K464 and M155" to "does not reproduce phenotypes similar to K464A or K464E, and therefore does not affect the function of K464" in the two instances where the phrase appears here.

In addition, on page 12 (lines 204-205), I suggest changing the sentence "The negatively charged residues E243, D244, and E247 on the S2-S3 linker are additional interaction partners of K464." to "We next checked for interactions of K464 with the negatively charged residues E243, D244, and E247 on the S2-S3 linker, which were also identified as potential interaction partners of K464 (Figure 1B)". This will create a stronger link between this paragraph and the preceding text.

In the revised DISCUSSION section (page 25, lines 489-504), I suggest changing the sentences "We next studied two negatively charged residues in the opposite S2-S3 linker, D244 and E247. Neither neutralization nor charge reversal at those positions resulted in phenotypes similar to K464A or K464E." to "We next studied two negatively charged residues in the opposite S2-S3 linker, D244 and E247, which had also been identified as potential interaction partners for K464. However, neither neutralization nor charge reversal at those positions resulted in phenotypes similar to K464A or K464E." This will create a stronger link between this paragraph and the preceding DISCUSSION text.

Finally, in the revised DISCUSSION section (page 25, lines 489-504), I suggest changing the sentence "This finding may further suggest that D244 or E247 take over the role of an interaction partner of K464 if the respective other residue is substituted." to "This finding may suggest that D244 or E247 take over the role of an interaction partner of K464 if the respective other residue is

substituted (as in the case of the D244 and E247 mutants).". The word "further" seems out of place here, and this revision would also clarify the last part of the sentence.

Comment #2: It should be pointed out that the smaller overall degree of structural shift (relative to the apo wild type mHCN2 MD simulations) that was observed for the wild type mHCN2 simulations with bound cAMP, compared to the apo K464E mHCN2 simulations (as exemplified in Figure 3B), may alternatively be due to inherent limitations of the unbiased MD simulation methodology. However, this does not seem to be acknowledged here.

Author Response: As suggested, we now include such a paragraph in the Discussion section (pages 24 and 25, lines 455-481), such that it reads: "One might argue that the small differences between the apo K464E mHCN2 simulations may be due to inherent limitations of the unbiased MD simulation. However, we followed the extensively validated "ensemble average approach", a procedure to interpret equilibrium dynamics from multiple, independent MD trajectories (Loccisano et al., 2004 (DOI: 10.1016/j.jmfm.2003.12.004); Likic et al., 2005 (DOI: 10.1110/ps.051681605); Caves et al., 2008 (DOI: 10.1002/pro.5560070314), to minimize the potential impact of insufficient sampling. Moreover, we started the MD simulations from the same channel conformation, allowing to interpret results between the mHCN2 simulations on a relative basis."

==> Reply to Author Response:
The comment has been addressed to my satisfaction.

Comment #3: For the Results subsection "K464 acts via backbone but not via side-chain interaction with M155 of the opposite subunit", the authors could have better emphasized that the steady-state activation relationship shifts observed for the M155R and M155E mutants were shifts in directions opposite to the directions that would have been expected if K464 interacted with the side chain of M155, thereby ruling out interaction with the side chain of M155 as a likely contributor to the observed channel gating behavior.

Author Response: We thank the reviewer for this comment and rephrased the respective paragraph in the RESULTS section (page 14, lines 262-269):
"Neither shortening the side-chain at position 155 (M155A) nor adding a positive charge (M155R) to test for repulsive forces between position M155 and K464 affected the gating behavior in a similar way as did K464E, which supports our idea that the side chain at position 155 is not interacting directly with the side chain of K464. Moreover, in M155E, analysis of steady-state activation revealed an open state-stabilization. If the introduced glutamate side chain interacted with the K464 side chain, most likely by forming a stabilizing salt bridge between HCN domain and opposite C-linker, we rather would expect a stabilization of the closed state."

==> Reply to Author Response:
The comment has been addressed to my satisfaction.

Reviewers' comments:

Reviewer #1 (Remarks to the Author):

I appreciate the effort of the Authors to improve the manuscript. But I regret to say that I still find the data interesting but overinterpreted with respect to a general mechanism in HCN channels. The proposed mechanism relies entirely on the phenotype of one mutation and the general mechanism is not supported by the data.

Comment #1

The Authors present their hypothesis that K464 makes a crucial interaction with M155 throughout the paper, but the data do not seem to support this picture:

Hypothesis of authors: The interaction of K464 with M155 is crucial for cAMP-mediated gating, in the sense that it is lost upon cAMP binding.

Test of the hypothesis: Arg464 was neutralized into Ala. This eliminates the crucial H-bond intramolecular interaction.

Results: The mutation has no effect on channel function and cAMP dependency.

My resume: the interaction between K464 and M155 cannot be as crucial as expected.

Authors' response to comment #1:

Please see the response for comment #3.

Comment #2

The authors then find that substitution of Arg464 with an anionic amino acid has an effect on the voltage dependency in that it shifts the activation curve to positive voltages.

Hypothesis of authors: The electro-repulsion, which is induced by this mutation, mimics the cAMP-bound conformation of the protein.

Test of the hypothesis: cAMP is added to the mutant channel. If the mutant would represent the cAMP-bound conformation of the protein, one would expect no further effect of cAMP.

Results: The data show that cAMP is still shifting the activation curve of the mutant, even more to the right.

My resume: the two effects are additive, i.e., the two mechanisms (that of the mutation and that of cAMP) are independent.

Authors' response to comment #2:

Please see the response for comment #3.

Comment #3

The authors analyze the effects of the K464E mutation by MD simulations of a homology model of HCN2.

Hypothesis of authors: They assume that the mutant conformation resembles that of the ligand bound protein. The expectation is that the apo-mutant protein and the holo-wt protein should behave the same.

Test of hypothesis: Comparison of the two proteins by MD shows that the ligand has no significant effect on the propensity of hydrogen bond formation between K464 and M155. Also, the ligand causes only a very small, if any, rotation of the CNBD. The mutation, on the other hand, generates a strong reduction in hydrogen bond formation and a visible albeit still small rotation of the CNBD.

Results: At this point, it should be mentioned that the 2.8° angle rotation is more similar to that of the structures of HCN1 (1° angle) that does not respond to cAMP, but very different from that of the hHCN4 (about 8°) that responds to cAMP.

My interpretation: the mutant is not reproducing the effect of the ligand. If anything, the mutant is evoking structural changes, which go beyond that of the ligand. But as we know from the functional data: the ligand is even adding a shift in the activation curve to the mutant. Hence the mutant cannot have a stronger effect than the ligand.

Authors' response to comment #3:

To clarify this issue, we report the charge inversion K464E mHCN2, which opens under less depolarized conditions, similar but not identical to the effect of cAMP binding.

We made several changes in the text, such that we now say that K464E, but not K464A, behaves similarly, yet not identical to the cAMP wild type channel. Such changes include parts of the Abstract, the Introduction (p. 4), and the Results (p. 7, 8, 10-12). In summary, the concluding paragraph (p. 8) now reads:

“In summary, the data show that K464E destabilizes the closed (autoinhibited) mHCN2 conformation, thereby affecting both the activation and the deactivation pathway. The similarity of the data obtained for wild type at saturating [cAMP] and K464E without cAMP led us to speculate that apo K464E and cAMP bound channels behave similarly, but both behave differently than the apo wild type channel. As K464A reveals only weak destabilizing properties on mHCN2, we assume that the destabilizing influence of K464E is likely mediated by charge interactions.”

Comment #4

Hypothesis: The authors conclude from the MD simulations that K464 interacts with M155A.

Test of hypothesis: Methionine has been mutated into Ala, Arg, and Glu.

Results: These mutations are not abolishing the cAMP dependency of the channel. The authors interpret this as evidence for an interaction of K464 with the amino acid backbone.

My interpretation: as a devil advocate, I could also conclude from these data that the results of the MD simulation are not validated and that K464 does not at all interact with M155. The absence of a functional effect is not a strong argument for the proposed interaction. This critical statement is further supported by the fact that the insertion of a cationic or anionic amino acid in this position has no impact on channel function. If we consider the long side chains of the cationic and anionic amino acids, it seems even surprising that they do not interact with the cationic K464.

Authors' response to comment #4:

We agree with the reviewer's suggestion that the absence of a functional effect alone is not necessarily a strong argument. Yet, there is a plausible explanation for why it is not surprising that cationic and anionic amino acids with long side chains very likely do not interact directly with K464. In the cryo-EM structure (PDB 5u6o), the side-chain of M155 is not pointing towards K464 but towards the potential location of the membrane.

We would also like to draw the reviewer's attention to that we assume that the main effect of K464E is also related to electrostatic interactions with neighboring residues, particularly those harbored on the S2/S3-linker. We are confident that the changes made throughout the manuscript now present this more clearly.

Comment #5

The authors find in the MD simulations that K464 does not only form H⁺ bonds with M155 but also, with a lower propensity, salt bridges with adjacent anionic amino acids.

Hypothesis: If the data from the MD simulation on H bond formation are correct also the salt bridge formations should be structurally relevant. This is in particular the case since salt bridges have in general a much higher energy than H-bonds.

Test of the hypothesis. Mutations, which eliminate the respective salt bridges were tested for cAMP sensitivity.

Result: the mutations have no appreciable effect on function.

My interpretation: it is at least possible that the predictions from the MD simulations are not correct. We have to consider here that the model is indeed only a homology model even though it is based on an experimentally resolved structure of the closely related HCN1 channel. But at this point it is important to remember that some of the structures, which are critical in this manuscript, such as the S2-S3 loop, are not even resolved properly in the HCN1 structure and they had to be modelled anew. They were modelled on the basis of general structural rules for proteins. This bears the hazard of artifacts in that the real structure may be different from the modeled one. To exemplify this, I noticed that the Authors used in one of the gap filling models

the cAMP bound structure of the CNBD from HCN1 to model the missing structures of the C and D helices in the apo. This seems to me a wrong maneuver since these domains undergo significant folding only after cAMP binding. This is way they are not present in the original apo structure of HCN1. Hence modeling this part of the apo HCN2 channel, guided by structures from the cAMP bound HCN1, could introduce serious artifacts.

Authors' response to comment #5:

We do not share the reviewer's concerns about the modeled S2/S3-loop and would like to use this opportunity to corroborate our opinion. Indeed, imprudent homology modeling may introduce artifacts, which, in turn, may lead to biased conclusions. In the present case, however, the modeled/predicted loop conformation aligns almost perfectly with the cryo-EM densities.

In Fig. 1, we show the superposition of our mHCN2 model (green), the hHCN1 structure (cyan; PDB 5u6o), and the reconstructed cryo-EM map (gray surface; EMD-8511), which was used to build an atomic hHCN1 structure de novo. For excellent visualization, the authors recommend a contour level (σ) of 0.04¹, at which the S2/S3-loop is not continuously visible, likely explaining why the authors did not model it. Still, this region is better visible at $\sigma = 0.035$, and even fully connected at $\sigma = 0.030$. In the latter case, our homology model aligns almost perfectly to the cryo-EM densities, with the side-chains pointing in the correct directions, showing that our homology model is in very good agreement with the cryo-EM data and, thus, likely not an artifact.

Still, a plausible explanation for why the S2/S3-loop is not permanently visible is that this region is most likely more flexible compared to the rest of the channel. Due to the higher number of conformations occupied by the S2/S3-loop, a single, high-resolution cryo-EM density is very unlikely, explaining the gap in the cryo-EM map at certain σ thresholds. Knowing that the S2/S3-loop is likely flexible in vitro, it is also plausible to assume that any effect mediated by the S2/S3-loop will not depend on a single, well-defined conformation. This may mediate the reviewer's concern about the initial conformation of the S2/S3-loop.

Fig. 1: Superposition of the mHCN2 model (green), the hHCN1 structure (cyan), and the reconstructed cryo-EM map (gray surface; EMD-8511) at three different contour levels. The arrow indicates breaks in the S2/S3 loop density at high resolution.

¹ For details, please visit: <https://www.ebi.ac.uk/emdb/EMD-8511?tab=overview>

Reviewer #2:

The authors addressed the reviewer's concerns adequately and incorporated additional explanations and discussion. I have no further comments.

Authors' response:

We appreciate the fruitful discussion and thank the reviewer for the previous suggestions.

Reviewer #3:

Review of Revised Manuscript COMMSBIO-20-2796A: Reply to Author

Revisions/Rebuttal of COMMSBIO-20-2796 Comments

Author Responses to COMMSBIO-20-2796 Comments:

Comment #1

The discussion within the Results subsection “Role of negatively charged residues in the S2-S3 loop for opposite subunit interactions” (pp. 13-14 of the article main text) was somewhat difficult to relate to the other results being presented, and possibly somewhat inconsistent with the corresponding text in the Discussion section. This Results subsection in its current state thus represents a possible weak point in the article.

Authors’ response from first revision:

We thank the reviewer for this comment and rephrased the paragraph “Role of negatively charged residues in the S2-S3 linker for opposite subunit interactions” in the RESULTS section (page 15,16, lines 272-293). It reads now:

“As described above, E247 in the S2-S3 linker was identified as a potential interaction partner for K464 (besides M155) to form a salt bridge. Thus, we constructed the mutants E247A and E247R to study the role of this residue for channel gating. The data are summarized in Figure 4B. Surprisingly, cAMP-induced gating is not affected. The cAMP-induced shift of $V_{1/2}$ is similar to the shift shown for wild type channels. However, there is a significant effect on voltage-induced gating: The steady-state activation relationships were shifted to more negative voltages for both mutants, indicating a stabilization of the closed state. From this, it can be concluded that changing the charge at position 247 in the S2-S3 linker, that way breaking a potential bond between E247 and K464, has no negative effect on the bond between K464 and M155. This result is further supported by the gating behavior of the mutant E247R_K464E. If a salt bridge between K464 and E247 mediated the function of K464, such a bond should be rescued in E247R_K464E, leading to a wild type-like phenotype. However, in E247R_K464E, the cAMP dependence was still strongly reduced like in K464E ($V_{1/2} = 4.2 \pm 1.2$ mV) (Figure 4B).

Additionally, our MD data showed a low probability of forming salt bridge interactions between K464 and D244 (Figure 3A). We tested the role of D244 for channel gating by constructing D244A and D244K. In both cases, $V_{1/2}$ values were shifted to more negative values in the absence of cAMP, while there were no changes for cAMP-saturated constructs (Figure 4C). Consequently, $\Delta V_{1/2}$ was significantly increased rather than decreased as shown for K464E. From this, it can be concluded that changing the charge at position 244 in the S2-S3 linker, that way breaking a potential bond between E244 and K464, has no negative effect on the bond between K464 and M155.”

We also adapted the respective paragraph in the DISCUSSION section (page 25, lines 489-504):

“We next studied two negatively charged residues in the opposite S2-S3 linker, D244 and E247. Neither neutralization nor charge reversal at those positions resulted in phenotypes similar to

K464A or K464E. We thus concluded that the function of K464 in mHCN2 wild type does not rely on the formation of a bond with either one of those residues. To further support this conclusion for the residue E247, which showed the highest likelihood of developing a salt bridge with K464, we confirmed with the mutant E247R_K464E that rescuing a possible salt bridge does not result in a wild type-like phenotype. This construct still behaved like the K464E mutant, possibly because the proposed repulsive forces introduced by the glutamate at position K464 still find another counterpart in other negatively charged residues near position 247, for instance, D244. This finding may further suggest that D244 or E247 take over the role of an interaction partner of K464 if the respective other residue is substituted.

Notably, all four constructs showed a shift of $V_{1/2}$ to more negative values in the absence of cAMP, indicating a stabilization of the closed state. Because $V_{1/2}$ at saturating cAMP is less affected, for all constructs, the cAMP-induced $\Delta V_{1/2}$ is more pronounced than for mHCN2 wild type. Thus, the negatively charged residues in the S2-S3 linker seem to be involved in voltage-dependent rather than in cAMP-dependent gating.”

==> Reply to Author Response:

In the revised “Role of negatively charged residues in the S2-S3 linker for opposite subunit interactions” RESULTS subsection (page 15,16, lines 272-293), the phrase “has no negative effect on the bond between K464 and M155” seems somewhat out of place here, although I understand that this text was also added to try to address comments from the other reviewers. I suggest changing the phrase “has no negative effect on the bond between K464 and M155” to “does not reproduce phenotypes similar to K464A or K464E, and therefore does not affect the function of K464” in the two instances where the phrase appears here.

In addition, on page 12 (lines 204-205), I suggest changing the sentence “The negatively charged residues E243, D244, and E247 on the S2-S3 linker are additional interaction partners of K464.” to “We next checked for interactions of K464 with the negatively charged residues E243, D244, and E247 on the S2-S3 linker, which were also identified as potential interaction partners of K464 (Figure 1B).”. This will create a stronger link between this paragraph and the preceding text.

In the revised DISCUSSION section (page 25, lines 489-504), I suggest changing the sentences “We next studied two negatively charged residues in the opposite S2-S3 linker, D244 and E247. Neither neutralization nor charge reversal at those positions resulted in phenotypes similar to K464A or K464E.” to “We next studied two negatively charged residues in the opposite S2-S3 linker, D244 and E247, which had also been identified as potential interaction partners for K464. However, neither neutralization nor charge reversal at those positions resulted in phenotypes similar to K464A or K464E.” This will create a stronger link between this paragraph and the preceding DISCUSSION text.

Finally, in the revised DISCUSSION section (page 25, lines 489-504), I suggest changing the sentence “This finding may further suggest that D244 or E247 take over the role of an interaction partner of K464 if the respective other residue is substituted.” to “This finding may suggest that D244 or E247 take over the role of an interaction partner of K464 if the respective other residue is substituted (as in the case of the D244 and E247 mutants).”. The word “further” seems out of place here, and this revision would also clarify the last part of the sentence.

Authors' response to comment #1:

We kindly appreciated the suggestions and included all suggestions in the revised version.

Comment #2

It should be pointed out that the smaller overall degree of structural shift (relative to the apo wild type mHCN2 MD simulations) that was observed for the wild type mHCN2 simulations with bound cAMP, compared to the apo K464E mHCN2 simulations (as exemplified in Figure 3B), may alternatively be due to inherent limitations of the unbiased MD simulation methodology. However, this does not seem to be acknowledged here.

Authors' response from first revision:

As suggested, we now include such a paragraph in the Discussion section (pages 24 and 25, lines 455-481), such that it reads: “One might argue that the small differences between the apo K464E mHCN2 simulations may be due to inherent limitations of the unbiased MD simulation. However, we followed the extensively validated “ensemble average approach”, a procedure to interpret equilibrium dynamics from multiple, independent MD trajectories (Loccisano et al., 2004 (DOI: 10.1016/j.jmngm.2003.12.004); Likic et al., 2005 (DOI: 10.1110/ps.051681605); Caves et al., 2008 (DOI: 10.1002/pro.5560070314), to minimize the potential impact of insufficient sampling. Moreover, we started the MD simulations from the same channel conformation, allowing to interpret results between the mHCN2 simulations on a relative basis.”

==> Reply to Author Response:

The comment has been addressed to my satisfaction.

Authors' response to comment #2:

Thank you.

Comment #3

For the Results subsection “K464 acts via backbone but not via side-chain interaction with M155 of the opposite subunit”, the authors could have better emphasized that the steady-state activation relationship shifts observed for the M155R and M155E mutants were shifts in directions opposite to the directions that would have been expected if K464 interacted with the side chain of M155, thereby ruling out interaction with the side chain of M155 as a likely contributor to the observed channel gating behavior.

Authors' response from first revision:

We thank the reviewer for this comment and rephrased the respective paragraph in the RESULTS section (page 14, lines 262-269):

“Neither shortening the side-chain at position 155 (M155A) nor adding a positive charge (M155R) to test for repulsive forces between position M155 and K464 affected the gating behavior in a similar way as did K464E, which supports our idea that the side chain at position 155 is not interacting directly with the side chain of K464. Moreover, in M155E, analysis of steady-state activation revealed an open state-stabilization. If the introduced glutamate side

chain interacted with the K464 side chain, most likely by forming a stabilizing salt bridge between HCN domain and opposite C-linker, we rather would expect a stabilization of the closed state.”

==> Reply to Author Response:

The comment has been addressed to my satisfaction.

Authors' response to comment #3:

We appreciated the fruitful discussion and thank the reviewer for the previous suggestions.